# A Revisit of Total Correlation in Disentangled Variational Auto-Encoder with Partial Disentanglement

## Abstract

A fully disentangled variational auto-encoder (VAE) aims to identify disentangled latent components from observations. However, enforcing full independence between all latent components may be too strict for certain datasets. In some cases, multiple factors may be entangled together in a non-separable manner, or a single independent semantic meaning could be represented by multiple latent components within a higher-dimensional manifold. To address such scenarios with greater flexibility, we propose the Partially Disentangled VAE (PDisVAE), which generalizes the total correlation (TC) term in fully disentangled VAEs to a partial correlation (PC) term. This framework can handle group-wise independence and can naturally reduce to either the standard VAE or the fully disentangled VAE. Validation through three synthetic experiments demonstrates the correctness and practicality of PDisVAE. When applied to real-world datasets, PDisVAE discovers valuable information that is difficult to find using fully disentangled VAEs, implying its versatility and effectiveness.

## 1 Introduction

Disentangling independent latent components from observations is a desirable goal in representational learning (Bengio et al., 2013; Alemi et al., 2016; Schmidhuber, 1992; Achille & Soatto, 2017), with numerous applications in fields such as computer vision and image processing (Lake et al., 2017), signal analysis (Hyvärinen & Oja, 2000; Hyvarinen & Morioka, 2017), and neuroscience (Zhou & Wei, 2020; Yang et al., 2021; Wang et al., 2024; Calhoun et al., 2009). To disentangle latent components in an unsupervised manner, most models employ techniques that combine optimizing a variational auto-encoder (VAE) (Kingma, 2013) with an additional penalty term known as total correlation (mutual information) (Kraskov et al., 2004), classified as fully disentangled VAEs (Higgins et al., 2017; Kim & Mnih, 2018; Chen et al., 2018).

Variants of fully disentangled VAEs include modifying the extra penalty term of the VAE loss function (Meo et al., 2024; Hsu et al., 2024), using auxiliary information to supervise the latent (Ahuja et al., 2022), or designing particular decoder structure (Bhowal et al.). These related works are summarized in Tab. 3 in Appendix. A.5. Regardless of the techniques employed in these works, they all result in the latent components being fully independent of each other.

However, enforcing full independence among all latent components can be an overly strong assumption for certain datasets. For instance, consider the location coordinates $(x, y)$ of a set of points in a 2D plane. If the points are uniformly distributed within a square $[-1, 1] \times [-1, 1]$, the location distribution can be expressed as $p(x, y) = p(x)p(y)$, indicating that $x$ and $y$ are independent components. However, if the points are distributed in an irregular shape, such as a butterfly, the $(x, y)$ coordinates become entangled, resulting in $p(x, y) \neq p(x)p(y)$. In this case, the location information cannot be decomposed into two independent components but must be jointly represented by $(x, y)$ together. If the points also have attributes independent of their location, such as RGB color represented by a 3D vector, we then encounter the **group-wise independence**, where a rank-2 entangled group (location) is independent of a rank-3 entangled group (color).

To deal with such group-wise independence, we propose the partially disentangled VAE (PDisVAE).
• First, it generalizes the total correlation (TC) penalty term in the loss function of fully disentangled VAEs to partial correlation (PC). PC explicitly penalizes group-wise independence while permitting within-group entanglement. This unified formulation of PC encompasses both the standard VAE

and fully disentangled VAEs.

• Second, we revisit the batch approximation method used for computing PC and TC. The existing batch approximation method proposed by Chen et al. (2018) for computing TC in fully disentangled VAEs exhibits a high variance in the estimator. Since accurate batch approximation is critical for the success of the method, we derive the optimal importance sampling (IS) batch approximation formula and provide a theoretical proof of its optimality.

## 2 BACKGROUNDS: FULLY DISENTANGLED VAEs

### 2.1 BY TOTAL CORRELATION (TC)

Given a dataset of observations $\left\{\boldsymbol{x}^{(n)}\right\}_{n=1}^{N}$ consisting of $N$ samples, fully disentangled VAEs aim to identify $K$ statistically independent (disentangled) latent components, $z_1 \perp \cdots \perp z_K$, within the latent variable $\boldsymbol{z} \in \mathbb{R}^K$ that generate the observation $\boldsymbol{x}$. To achieve full disentanglement, fully disentangled VAEs optimize the following objective function (Blei et al., 2017):

$$\mathcal{L} = \frac{1}{N} \sum_{n=1}^{N} \left[ \underbrace{\left( \mathbb{E}_{q(\boldsymbol{z}|\boldsymbol{x})} \left[ \ln p\left(\boldsymbol{x}^{(n)} \middle| \boldsymbol{z}\right) \right] \right)}_{\text{reconstruction log-likelihood}} - \underbrace{\text{KL}\left( q\left(\boldsymbol{z}|\boldsymbol{x}^{(n)}\right) \middle\| p(\boldsymbol{z}) \right)}_{\text{KL divergence}} \right] - \beta \cdot \underbrace{\text{KL}\left( q(\boldsymbol{z}) \middle\| \prod_{k=1}^{K} q(z_k) \right)}_{\text{total correlation (TC)}}. \quad (1)$$

In the function, $p(\boldsymbol{x}|\boldsymbol{z};\theta) = \mathcal{N}\left(\boldsymbol{x}; \boldsymbol{\mu}, \boldsymbol{\sigma}^2\right)$, $\boldsymbol{\mu}, \boldsymbol{\sigma}^2 = \text{decoder}(\boldsymbol{z};\theta)$, where $\text{decoder} : \mathbb{R}^K \rightarrow \mathbb{R}^D$ is parameterized by $\theta$. $q(\boldsymbol{z}|\boldsymbol{x};\phi) = \mathcal{N}(\boldsymbol{z}; \boldsymbol{\mu} =, \boldsymbol{\sigma}^2)$, $\boldsymbol{\mu}, \boldsymbol{\sigma}^2 = \text{encoder}(\boldsymbol{x};\theta)$ is the variational distribution, in which the $\text{encoder} : \mathbb{R}^D \rightarrow \mathbb{R}^K$ is parameterized by $\phi$. In Eq. (1) and the following, we omit $\theta$ in $p$ and $\phi$ in $q$ for simplification without loss of clarity. The prior $p(\boldsymbol{z})$ is often chosen to be a standard normal prior. Standard VAE consists of the first two terms. The last term is the total correlation (TC), where $q(\boldsymbol{z}) = \frac{1}{N} \sum_{n=1}^{N} q(\boldsymbol{z}, n) = \sum_{n=1}^{N} q\left(\boldsymbol{z}|\boldsymbol{x}^{(n)}\right) q(n)$ is the aggregated posterior, followed by Makhzani et al. (2015). $q(\boldsymbol{z})$ can be viewed as a Gaussian kernel density estimation from $\left\{\boldsymbol{z}^n\right\}_{n=1}^{N}$ in latent space. The goal of this TC term is to achieve $q(\boldsymbol{z}) = \prod_{k=1}^{K} q(z_k)$, which is the rigorous definition of independence among $z_1, ..., z_k$. That is why Eq. (1) can achieve full disentanglement compared with standard VAE.

Before the development of the disentangled VAE in Eq. (1), Higgins et al. (2017) and Burgess et al. (2018) initially discovered that penalizing the entire KL divergence term in the standard VAE can increase the latent disentanglement. So, In their $\beta$-VAE, there is no TC term, but rather a penalty coefficient on the KL divergence term. It was found later by Kim & Mnih (2018) and Chen et al. (2018) and summarized by Dubois et al. (2019) that the effective term for enhancing the latent disentanglement is indeed the TC. Consequently, they developed Eq. (1) with $\beta > 0$, resulting in FactorVAE and $\beta$-TCVAE.

### 2.2 BY A NON-GAUSSIAN PRIOR (ICA)

Another approach to achieving full disentanglement is to view the problem as an independent component analysis (ICA). The core idea inspired by ICA is that "non-Gaussian is independent" (Hyvärinen & Oja, 2000; Hyvärinen et al., 2009). In short, we need to assume $p(\boldsymbol{z})$ to be non-Gaussian. The logcosh distribution is one of the most commonly used:

$$p(\boldsymbol{z}) = p(z_1, \ldots, z_K) = \prod_{k=1}^{K} p(z_k) = \prod_{k=1}^{K} \frac{\pi \left( \text{sech} \frac{\pi z_k}{2\sqrt{3}} \right)^2}{4\sqrt{3}}, \quad (2)$$

where $\text{sech} = \frac{1}{\cosh}$ is the hyperbolic secant function.

In traditional linear ICA, $\boldsymbol{x} = \boldsymbol{f}(\boldsymbol{z})$ where $\boldsymbol{f} : \mathbb{R}^K \rightarrow \mathbb{R}^D$ is a full-rank ($D = K$) linear deterministic mapping, and $p(\boldsymbol{x}|\boldsymbol{z}; \boldsymbol{f}) = \delta(\boldsymbol{x} - \boldsymbol{f}(\boldsymbol{z}))$ ($\delta$ is the Dirac delta function), then we can use maximum likelihood estimate (MLE) to learn $\boldsymbol{f}$ via the "change of variable" formula,

$$p(\boldsymbol{x}) = \int p(\boldsymbol{x}|\boldsymbol{z}; \boldsymbol{f}) p(\boldsymbol{z}) \, \text{d}\boldsymbol{z} = \left| \det \frac{\text{d}\boldsymbol{f}^{-1}}{\text{d}\boldsymbol{z}} \right| \cdot p(\boldsymbol{f}^{-1}(\boldsymbol{x})), \quad (3)$$

and recover $\boldsymbol{z} = \boldsymbol{f}^{-1}(\boldsymbol{x})$. However, there are two main drawbacks to this traditional linear ICA. First, it cannot be extended to non-invertible non-linear $\boldsymbol{f}(\boldsymbol{z})$ since the $\left| \det \frac{\text{d}\boldsymbol{f}^{-1}}{\text{d}\boldsymbol{z}} \right|$ in the "change of variable" formula becomes intractable (Khemakhem et al., 2020; Sorrenson et al., 2020). Second, observation $\boldsymbol{x} \in \mathbb{R}^D$ is usually in higher dimensional space than $\boldsymbol{z}$ ($D > K$) with noises, which are not explicitly modeled by traditional linear ICA.

To address these issues, we use a VAE with a logcosh prior $p(\boldsymbol{z})$ defined in Eq. (2). It is worth mentioning that, to the best of our knowledge, we are the first to recognize the logcosh-priored VAE as the nonlinear ICA problem. However, certain limitations remain. For instance, if the true number of disentangled latent components is two but we instruct the logcosh-priored VAE to find three, it will yield three components with poor disentanglement instead of finding two disentangled components and one non-informative component. We will discuss this limitation in detail in the experiment section. Additionally, the logcosh-priored VAE cannot be extended to a partially disentangled version, since the logcosh prior does not support partial independence.

## 3 PARTIALLY DISENTANGLED VAE (PDISVAE)

### 3.1 PROBLEM DEFINITION

Although several approaches have been introduced in Sec. 2, a common issue among them is they are all trying to find "fully disentangled (independent)" latent space. However, if the true latent variables are partially disentangled by groups, applying a fully disentangled method is hard to successfully recover the underlying latent structure accurately.

We first formally define partial disentanglement (independence). Still, assume latent $\boldsymbol{z} \in \mathbb{R}^K$, but now the latent dimensions are disentangled by $G$ groups, while each group has its internal within-group rank $H$, satisfying $K = G \times H$. For simplicity, we also denote the $g$-th group as $\boldsymbol{z}_g = (z_{(g-1)H+1}, \ldots, z_{gH})$, so that $\boldsymbol{z} = (\boldsymbol{z}_1, \ldots, \boldsymbol{z}_G)$. For simplicity and without loss of generality, we assume all groups have the same group rank $H$, but this requirement can be easily relaxed to arbitrary group ranks in different groups. Then, the **partially disentangled** latent can be formulated as

$$(z_1, \ldots, z_H) \perp (z_{H+1}, \ldots, z_{2H}) \perp \cdots \perp (z_{K-H+1}, \ldots, z_K) \iff p(\boldsymbol{z}) = \prod_{g=1}^{G} p(\boldsymbol{z}_g). \quad (4)$$

This equation expresses that within each group, latent components may exhibit dependencies and may not be further disentangled. However, the groups themselves remain independent of each other. We refer to this as **group-wise independence**. For example, when $K = 6$ and there are $G = 3$ groups, the three groups are independent of each other as $(z_1, z_2) \perp (z_3, z_4) \perp (z_5, z_6) \iff p(z_1, \ldots, z_6) = p(z_1, z_2)p(z_3, z_4)p(z_5, z_6)$, while dimensions within each group can be highly dependent and cannot be further decomposed, i.e., $p(z_1, z_2) \neq p(z_1)p(z_2)$, $p(z_3, z_4) \neq p(z_3)p(z_4)$, $p(z_5, z_6) \neq p(z_5)p(z_6)$.

### 3.2 PARTIAL CORRELATION

To identify partially independent component groups as defined above, one might consider a straightforward approach: using existing methods to impose marginal independence on between-group components. For instance, if we have $(z_1, z_2) \perp z_3$, one might attempt to apply existing algorithms to require $z_1 \perp z_3$ and $z_2 \perp z_3$. However, this is generally NOT correct. Specifically, the former is a sufficient but not necessary condition ($\implies$) for the latter. A simple counterexample is the distribution $p(z_1, z_2, z_3)$, where $p(0, 0, 1) = p(0, 1, 0) = p(1, 0, 0) = p(1, 1, 1) = 0.25$. It can be verified that $(z_1, z_2) \not\perp z_3$, while $z_1 \perp z_3$ and $z_2 \perp z_3$. For more detailed explanations regarding marginal independence and group-wise independence, please refer to Appendix A.1. Therefore, we must explicitly enforce $(z_1, z_2) \perp z_3$.

To explicitly require group-wise independence, we introduce the **partially disentangled VAE (PDisVAE)**, which replaces the TC in Eq. (1) with a partial correlation (PC) term. Specifically, given a dataset of $N$ equally treated samples, the probability of taking the $n$-th sample is $q(n) = \frac{1}{N}$, so that $\frac{1}{N} \sum_{n=1}^{N}[\cdot] = \mathbb{E}_{q(n)}[\cdot]$. Also let $q(\boldsymbol{z}|n) \coloneqq q(\boldsymbol{z}|\boldsymbol{x}^{(n)})$. Then, the target function to be maximized when observing the whole dataset $\{\boldsymbol{x}^{(n)}\}_{n=1}^{N}$ is

$$\mathcal{L} = \frac{1}{N} \sum_{n=1}^{N} \left[ \underbrace{\left( \mathbb{E}_{q(\boldsymbol{z}|\boldsymbol{x})} \left[ \ln p\left(\boldsymbol{x}^{(n)}\big|\boldsymbol{z}\right) \right] \right)}_{\text{reconstruction log-likelihood}} - \underbrace{\mathrm{KL}\left( q\left(\boldsymbol{z}\big|\boldsymbol{x}^{(n)}\right) \big\| p(\boldsymbol{z}) \right)}_{\text{KL divergence}} \right] - \underbrace{\beta \cdot \mathrm{KL}\left( q(\boldsymbol{z}) \bigg\| \prod_{g=1}^{G} q(\boldsymbol{z}_g) \right)}_{\text{partial correlation (PC)}}. \quad (5)$$

The partial correlation (PC) term is responsible for disentangling independent groups. When $q(\boldsymbol{z}) = \prod_{g=1}^{G} q(\boldsymbol{z}_g)$, $\mathrm{KL}\left( q(\boldsymbol{z}) \big\| \prod_{g=1}^{G} q(\boldsymbol{z}_g) \right) = 0$. Otherwise, this PC term is greater than 0 and is penalized by $\beta$, as the hyperparameter $\beta > 0$.

It is worth noting that when $G = 1$, $\mathrm{PC} \equiv 0$ and Eq. (5) becomes the standard VAE objective function; when $G = K$, PC is just the total correlation (TC) and Eq. (5) becomes the FactorVAE (Kim

& Mnih, 2018) and/or $\beta$-TCVAE objective function (Chen et al., 2018). Appendix. A.2 contains a detailed derivation of the ELBO decomposition of Eq. (5).

### 3.3 THE BEHAVIOR OF PDisVAE

In the previous subsection, we introduced PDisVAE to identify group-wise independence but did not discuss what to expect within the groups discovered by PDisVAE. Here, we will outline three potential relationships that the latent components within a group could exhibit. To illustrate, let us consider a discovered latent pair $(\hat{z}_i, \hat{z}_j)$; the three cases of interest are illustrated in Fig. 1.

• **Case 1: Non-separable dependent.** Consider we have the true latent $(z_i, z_j)$ from the equations shown in the right plot of case 1, where both the mean and variance of the Gaussian $z_j$ are dependent on $z_i$. This makes $z_i$ and $z_j$ highly entangled with each other in one group and it is impossible to further separate them independently by any linear transformation. Then, PDisVAE should identify a group $(\hat{z}_i, \hat{z}_j)$ that cannot be further separated independently through any linear transformation. Furthermore, we should be able to align the estimated $(\hat{z}_i, \hat{z}_j)$ with the true $(z_i, z_j)$ via a linear transformation. In this case, the within-group TC cannot become zero under any linear transformation.

• **Case 2: Rank-deficient.** Consider that PDisVAE has identified an estimated group $(\hat{z}_i, \hat{z}_j)$ in the left plot of case 2. Although they are dependent, they exhibit a clear linear relationship, which means they can be reduced to a single effective component, $z_i$, while $z_j$ serves as a dummy latent component. For example, if we have three latent components such that $(z_1, z_2) \perp z_3$, and we apply PDisVAE with $K = 4 = G \times H = 2 \times 2$, we would expect to find a dummy component $z_4 \approx 0$ in the second group, resulting in $(z_1, z_2) \perp (z_3, z_4 \approx 0)$. To verify the presence of a dummy latent, one could apply principal component analysis (PCA) to the group and identify a significantly small principal component, or conduct a normality test to detect Gaussian noise.

• **Case 3: Independent.** In this example, $\hat{z}_i$ and $\hat{z}_j$ are irreducibly dependent on each other. However, it is possible to further separate them into independent components via a linear transformation, resulting in the right plot that $z_i$ and $z_j$ become uniform distributions independent of each other. Consequently, $\hat{z}_i$ and $\hat{z}_j$ identified by PDisVAE should be allocated to two different groups rather than the same group. In this case, the within-group TC can be reduced to zero after a particular linear transformation. This indicates that as long as PDisVAE accurately identifies enough independent groups, the latent components within each group should not be independent of one another.

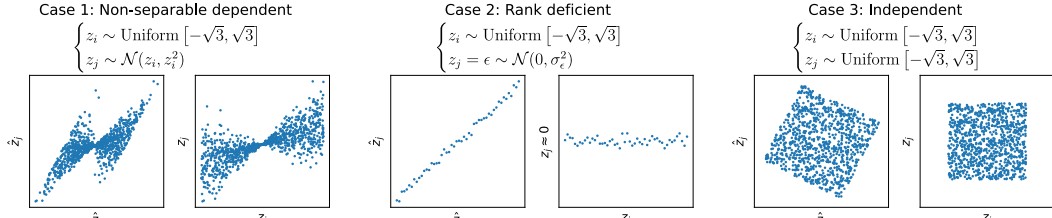

Figure 1: Visual illustrations for the desired behavior of the PDisVAE. In each case, the left plot is the estimated latent $(\hat{z}_i, \hat{z}_j)$ and the right plot is the true latent $(z_i, z_j)$.

### 3.4 BATCH APPROXIMATION

During training, strictly computing the aggregated joint/group posterior of the form $q(z) = \sum_{n=1}^{N} q(z|n)q(n) = \frac{1}{N} \sum_{n=1}^{N} q(z|n)$ might be unfeasible, since we only have a batch of size $M$, denoted as $\mathcal{B}_M := \{n_1, n_2, \ldots, n_M\}$ without replacement. Although Chen et al. (2018) proposed two approximation methods (the top two methods in Tab. 1), we argue that our **importance sampling (IS)** derived in the following paragraph and compared in Tab. 1 is a better approach.

Intuitively, when we only have a batch $\mathcal{B}_M \subsetneq \{1, \ldots, N\}$ and a sampled $z \sim q(z|n_*)$, where $n_*$ is a specific example point in $\mathcal{B}_M$, $q(z|n_*)$ is more likely to be greater than $q(z|n \neq n_*)$ since $z$ is sampled from $q(z|n_*)$. Therefore, we want the remaining $M - 1$ points in $\mathcal{B}_M \setminus \{n_*\}$ to represents the entire dataset excluding $n_*$, i.e., $\{1, 2, \ldots, N\} \setminus \{n_*\}$. Hence, an approximation of $q(z)$ at $z \sim q(z|n_*)$ could be

$$\hat{q}(z) = \frac{1}{N} q(z|n_*) + \sum_{n_m \in (\mathcal{B}_M \setminus \{n_*\})} \frac{N-1}{M-1} \frac{1}{N} q(z|n_m). \qquad (6)$$

Since each $q(z)$ is approximated using data points within a batch, it might be beneficial to shuffle the dataset every epoch to change the batch samples. Appendix. A.3 includes the complete derivation of this approximation, explaining why it is called IS approximation and proving its optimality, and an empirical evaluation of the three estimators. Notably, IS is more stable than MSS, as indicated by the relationship $\text{Var}[\text{IS}] < \text{Var}[\text{MSS}]$. Appendix. A.3 also proves the properties outlined in Tab. 1.

Table 1: Comparison of three batch approximation approaches. See Appendix. A.3 for more details.

|  | mean | variance |
|---|---|---|
| minibatch weighted sampling (MWS) | biased | |
| minibatch stratified sampling (MSS) | unbiased | $\text{Var}[\text{MSS}]$ in Eq. 20 |
| importance sampling (IS) | unbiased | $\text{Var}[\text{IS}] = \text{Var}[\text{MSS}] - \frac{M-2}{M(M-1)}$ in Eq. 21 |

## 4 EXPERIMENTS

**Alternative methods for comparison.** For evaluating the proposed PDisVAE, we compare it with the following three baselines:
• Standard **VAE** (Kingma, 2013): Theoretically, standard VAE does not have disentaglement ability.
• **ICA**: This is the logcosh-priored VAE for doing non-linear generative ICA inspired by Hyvärinen & Oja (2000).
• $\beta$-**TCVAE** (Chen et al., 2018): This method penalizes an extra total correlation (TC) term to achieve full disentanglement. It is theoretically equivalent to FactorVAE (Kim & Mnih, 2018).
• **PDisVAE**: Our proposed method penalizes the partial correlation (PC) term to achieve partial disentanglement. This is the only general method that can deal with group-wise independent latent. It reduces to the standard VAE when the number of groups $G = 1$; and reduces to the fully disentangled VAE when $G = K$, i.e., the number of groups equals the latent dimensionality.

We will begin by using two synthetic datasets to understand and rigorously validate the PDisVAE. Then, we will apply these methods to pdsprites, face images (CelebA), and neural data.

### 4.1 SYNTHETIC VALIDATION CASE 1: GROUP-WISE INDEPENDENT

**Dataset.** To validate that only PDisVAE is capable of dealing with group-wise independent datasets, we create a dataset consisting of $N = 2000$ points in $K = 6$ latent space $\boldsymbol{z}^{(n)} \in \mathbb{R}^6$, where three groups are independent of each other $(z_1, z_2) \perp (z_3, z_4) \perp (z_5, z_6)$, but components within each group are highly entangled. The three groups and their corresponding latent distributions are illustrated in Fig. 2(a). The observations $\boldsymbol{x}$ are linearly mapped from the latents $\boldsymbol{z}$ to a $D = 20$ dimensional space $\boldsymbol{x}^{(n)} \in \mathbb{R}^{20}$, and then Gaussian noise $\epsilon_d^{(n)} \overset{i.i.d.}{\sim} \mathcal{N}\left(0, 0.5^2\right)$ is added.

**Experimental setup.** For each method, we use Adam (Kingma, 2014) to train a linear encoder and a linear decoder (since the true generative process is linear) for 5,000 epochs. The learning rate is $5 \times 10^{-4}$ and the batch size is 128. For $\beta$-TCVAE and PDisVAE, the TC/PC penalty is set as $\beta = 4$. This is supported by Dubois et al. (2019), the $\beta$ selection in $\beta$-TCVAE (Chen et al., 2018), and our cross-validation result (Fig. 4) in the ablation study. Each method is run 10 times with different random seeds to evaluate and report its performance.

**Results.** The PC box plot in Fig. 2(b) shows that PDisVAE achieves the lowest partial correlation, implying that PDisVAE achieves the goal of disentangling latent in groups the best. Since this is the synthetic dataset and a model match experiment, we can align and match the estimated latent groups to their corresponding true latent groups to further validate the correctness of the latent estimation. The reconstruction $R^2$ of all four methods is approximately 0.97, indicating that all methods can reconstruct the observation perfectly. However, their learned latent representations are different. The latent $R^2$ in Fig. 2(b) shows that PDisVAE recovers the latent more accurately than others. Among the alternative, $\beta$-TCVAE is better than ICA and better than VAE. Since there is no independence assumption in standard VAE, it cannot recover the latent accurately. Fig. 3 visually shows that after aligning and matching with the true latent, PDisVAE recovers the latent best, which is consistent with the latent $R^2$ plot in Fig. 2(b).

An immediate question that arises is, how to check within-group latent estimated by PDisVAE is truly highly entangled and cannot be further decomposed, especially when there is no true latent. Essentially we hope to find case 1 within a group, rather than case 2 or case 3 illustrated in Fig. 1. The minimum within-group TC shown in Fig. 2(c) are all greater than 0, which means we indeed find highly entangled groups that cannot be further decomposed. Compared to the minimum within-group TC, the close-to-zero pair TC between groups also indicates that components between groups are independent.

**Ablation.** To analyze the choice of the penalty coefficient $\beta$ of PC term in Eq. (5), we vary $\beta$ in PDisVAE from 0.1 to 100 and plot the cross validation results in Fig. 4. The PC and latent $R^2$

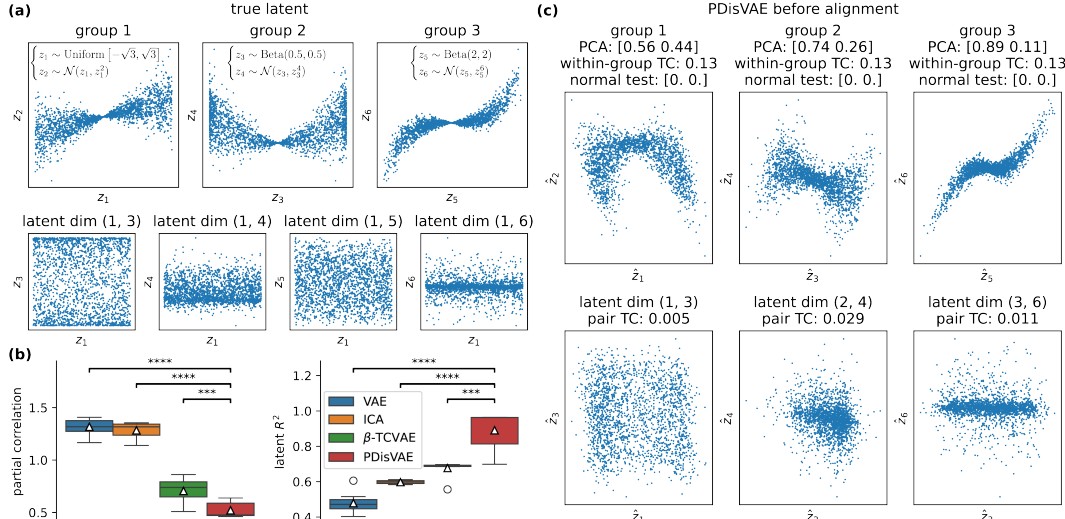

Figure 2: **(a)**: The true latent $z \in \mathbb{R}^6$ where three groups are $(z_1, z_2) \perp (z_3, z_4) \perp (z_5, z_6)$, but within-groups are highly entangled. **(b)**: The PC of the estimated latent and the latent $R^2$ after alignment to the true latent in (a). The $t$-test between PDisVAE and others shows that PDisVAE is significantly better than others (***: $p \leqslant 0.001$, ****: $p \leqslant 0.0001$). **(c)**: The estimated latent of PDisVAE before aligning to the true latent in (a). In each group, PCA shows the explained variance ratio in the group. Within-group TC shows the minimum TC under all possible linear transformations. The normal test shows the $p$-values of the null hypothesis that a marginal distribution is a normal distribution. If $p > 0.05$ for example, we may accept the null hypothesis that there exists a Gaussian noise dummy latent component. The pair TC is directly measured from the components in different groups.

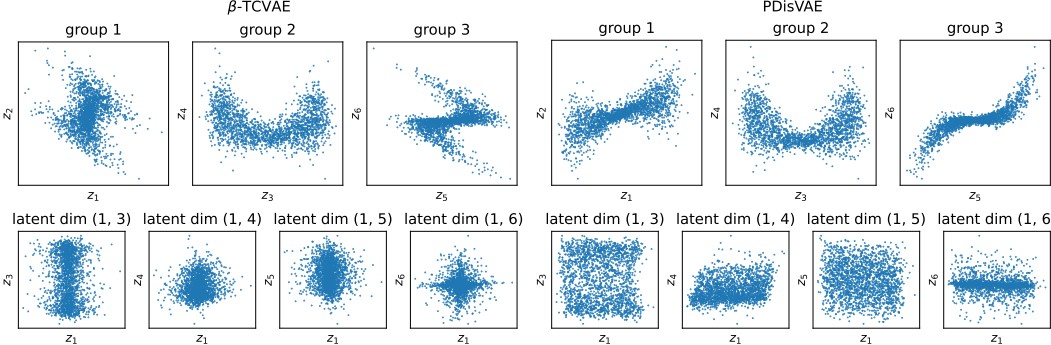

Figure 3: Latent alignment results for various methods, with each group aligned to the true latent from Fig.2(a). Some between-group pairs are also plotted to visually understand the marginally independent distributions between groups. VAE and ICA results are in Fig. 12 in Appendix. A.4.

plots indicate that $\beta > 1$ is necessary for an accurate recovery and effective minimization of the PC. However, excessively large $\beta$ might negatively impact reconstruction, as shown in the reconstruction $R^2$ plot. Hence, we recommend $\beta \in (2, 10)$, which supports our choice of $\beta = 4$ in our experiments.

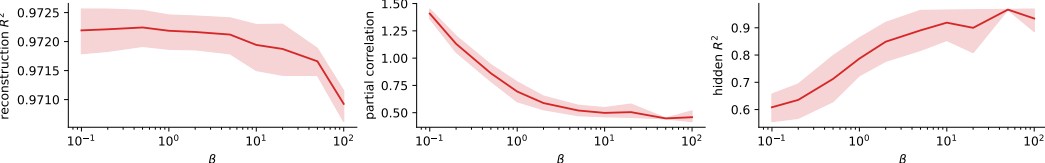

Figure 4: The cross-validation reconstruction $R^2$, PC, and latent $R^2$ w.r.t. the PC coefficient $\beta$.

### 4.2 SYNTHETIC VALIDATION CASE 2: FULLY INDEPENDENT

**Dataset and experimental setup.** To validate that PDisVAE can get the same results as from a fully disentangled VAE when the latent is fully independent, we create a dataset consisting of $N = 2000$ points in $K = 3$ latent space $z^{(n)} \in \mathbb{R}^3$, where the three latent components are inde-

pendent with each other $z_1 \perp z_2 \perp z_3$. Their distributions are shown in Fig. 5(a) and Fig. 6. The observation $\boldsymbol{x}$ is linearly mapped from the latent $\boldsymbol{z}$ to a $D = 20$ dimensional space $\boldsymbol{x}^{(n)} \in \mathbb{R}^{20}$, and then Gaussian noise $\epsilon_d^{(n)} \overset{i.i.d.}{\sim} \mathcal{N}\left(0, 0.5^2\right)$ are added. Although we only have $K = 3$ true latent components, we still learn $K = 6$ components to compare their flexibility when the true number of latent components is unknown. The experimental setup is the same as the previous one.

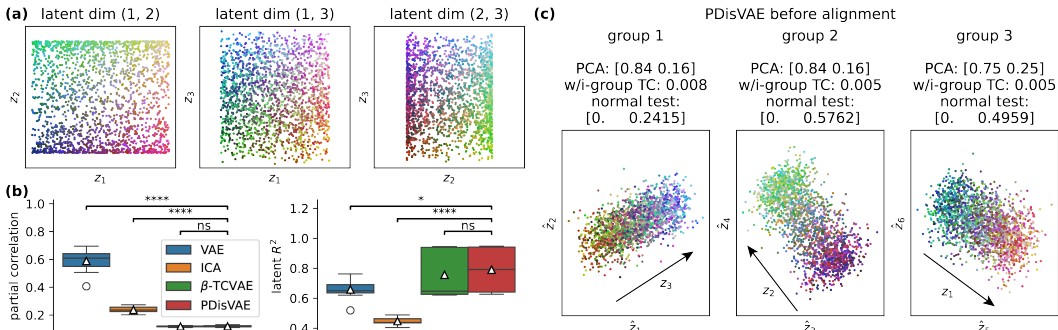

Figure 5: **(a)**: The true latent $\boldsymbol{z} \in \mathbb{R}^3$ coded by RGB $= z_1 z_2 z_3$, where three components are $z_1 \perp z_2 \perp z_3$. **(b)**: The PC of the estimated latent and the latent $R^2$ after alignment to the true latent in (a). The $t$-test between PDisVAE and others shows that PDisVAE is similar to $\beta$-TCVAE (ns: $p > 0.5$, *: $p \leqslant 0.05$, ****: $p \leqslant 0.0001$). **(c)**: The estimated latent of PDisVAE before aligning to the true latent shown in (a). The arrow in each plot shows the embedded true latent direction.

**Results.** The PC box plot and latent $R^2$ plot in Fig. 5(b) show that both $\beta$-TCVAE and PDisVAE achieve the lowest partial correlation and the highest latent $R^2$ on this fully disentangled dataset, which implies that PDisVAE automatically reduces to fully independent result if the group rank is deficient, as illustrated in case 2 in Fig. 1. In general, the actual group rank can be detected by PDisVAE and if the true group rank is less than the specified group dimensionality, dummy estimated latents will complemented in the corresponding group. Due to the strong requirement in ICA that tries to find logcosh-independent components but only three exist, ICA is not able to correctly identify three and find three dummy dimensions. This means logcosh might be too strong to allow the existence of dummy variables, which could be harmful when we do not know the true number of latent components. Fig. 6 also visually shows that $\beta$-TCVAE and PDisVAE accurately estimate the three latent distributions the best, which is consistent with the latent $R^2$ plot in Fig. 5(b).

To identify the three dummy latent dimensions complementing the three groups respectively through an unsupervised approach, we plot the PDisVAE result before alignment in Fig. 5(c). First, within-group TCs are all very small, indicating that the result is not the case 1 in Fig. 1. Since "independence is non-Gaussian", we can find a direction within each group that yields $p > 0.05$, which accepts the null hypothesis of the normal test that a Gaussian noise dummy dimension exists, corresponding to case 2 in Fig. 1. The arrows in Fig. 5(c) also visually indicate the embedded true latent direction.

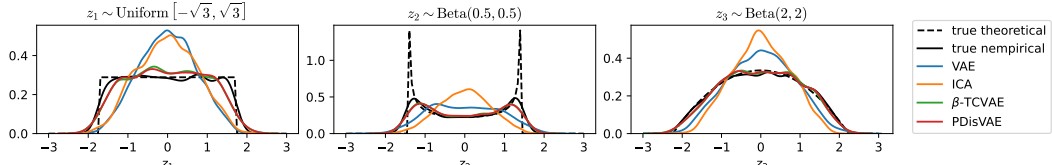

Figure 6: Estimated and true latent distribution after alignment to the true latent shown in Fig. 5(a).

### 4.3 SYNTHETIC APPLICATION: PARTIAL DSPRITES

**Dataset.** To understand the application scenario of PDisVAE, we created a synthetic dataset called partial dsprites (pdsprites), inspired by Matthey et al. (2017). Unlike the original dsprites, which features six fully independent latent dimensions, we only keep three latent components: $x$-location ($z_1$), $y$-location ($z_2$), and size ($z_3$), where $x$ and $y$ locations are entangled (not independent) with each other while this group is independent to the size, i.e., $(z_1, z_2) \perp z_3$. The generating process is depicted in Fig. 7(a), resulting in 805 gray-scaled images of shape $32 \times 32$.

**Experimental setup.** For each method, we use Adam to train a deep CNN VAE (Burgess et al., 2018) for 5,000 epochs with a learning rate of $1 \times 10^{-3}$. For $\beta$-TCVAE and PDisVAE, the TC/PC coefficient is set as $\beta = 4$. Given the true latent is $(z_1, z_2) \perp z_3$, learning two rank-2 groups

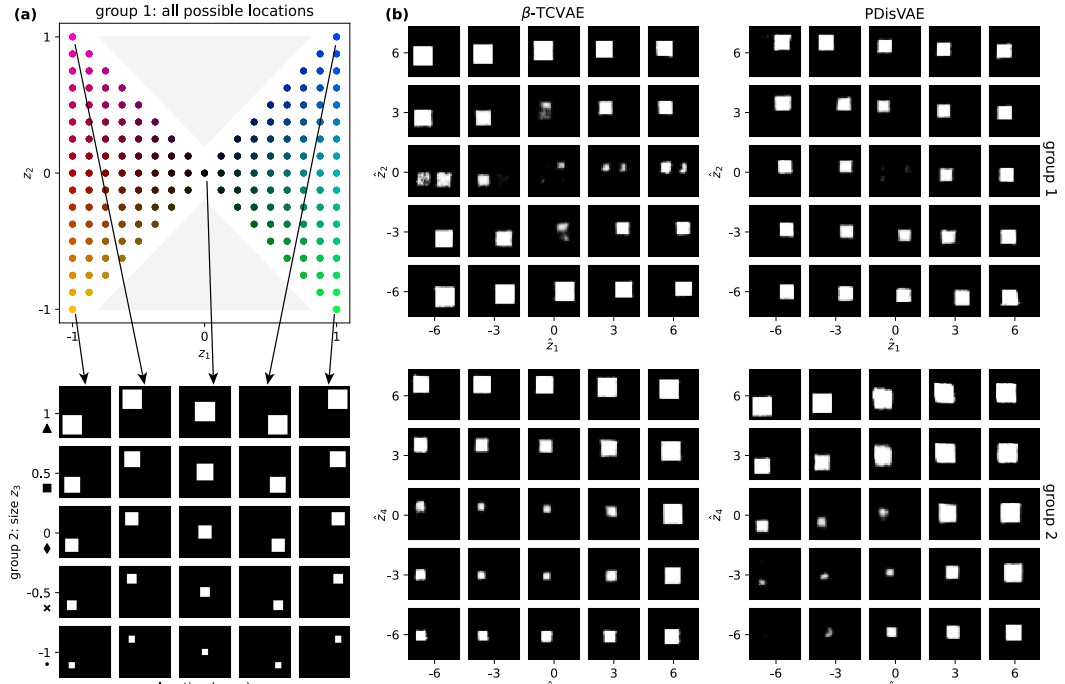

Figure 7: **(a)**: Latent and observation generating process. Locations $(z_1, z_2)$ are entangled, and uniformly distributed in a restricted region. Color represents the location information, with the upper and lower gray triangular areas being empty. The size $z_3$ is evenly distributed across five scales, represented by different markers, and is independent of the location. **(b)**: The reconstructed images by varying one of the latent groups $((\hat{z}_1, \hat{z}_2)$ or $(\hat{z}_3, \hat{z}_4))$ found by $\beta$-TCVAE and PDisVAE.

$(K = 4 = G \times H = 2 \times 2)$ should be able to find one group representing the location of the square and another rank-deficient group (contains a dummy latent component) representing the size of the square. Note that this setup is a model mismatch case, as we do not know the exact observation generating function $\boldsymbol{f}$; we only understand the semantic relationship between $\boldsymbol{z}$ and $\boldsymbol{x}$.

**Results.** Fig. 8 shows the estimated latent from all methods after alignment. PDisVAE has the highest latent $R^2$ and the lowest PC. Notably, PDisVAE successfully discovers two empty areas in the upper and lower gray triangular regions in group 1, reflecting the true latent distribution depicted in Fig. 7(a). Additionally, PDisVAE captures leveled size scales in $z_3$, showing smaller sizes for smaller $z_3$ and larger sizes for larger $z_3$, making it the closest representation of the true $z_3$ compared to other methods. Fig. 13 in Appendix. A.4 also plots the latent pair between groups.

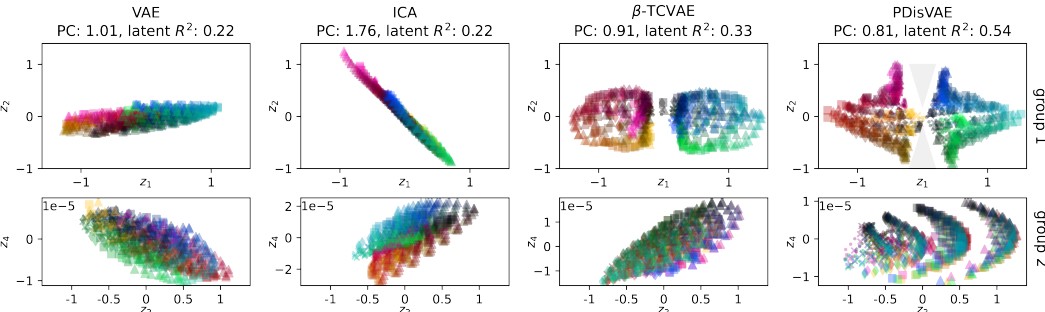

Figure 8: The latent plot after alignment for the group 1 $(z_1, z_2)$ and group 2 $(z_3, z_4 \approx 0)$ from different methods, and their corresponding PC and latent $R^2$. The color representation for location is the same as the color representation in Fig. 7(a), and the marker of the point in the latent plots represents the size of the square in the observation images.

Fig. 7(b) shows the reconstructed images by varying each of the two groups found by $\beta$-BTCVAE and PDisVAE, respectively. Group 1 from PDisVAE represents the location, with an empty center due to fewer observation samples in that area (see the region around $(z_1, z_2) = (0, 0)$ in Fig. 7(a)).

Besides, the square is expected to not appear in the top middle or bottom middle of the image, since there is no observation in the dataset that appears in those regions. The size is embedded in group 2, roughly along the $\hat{z}_4$ direction. In contrast, $\beta$-TCVAE mixes size and location in both groups because it enforces independence across all four components, which is incompatible with the fact that two location components are entangled together and independent of the third size component.

### 4.4 REAL-WORLD APPLICATIONS

To evaluate the performance and flexibility of PDisVAE in real-world applications, we train it on two real-world datasets, described in the following paragraphs. Since the true latent structure is unknown in these cases, we experiment with different group configurations for PDisVAE. Note that when $G = 1$, PC $\equiv 0$ and PDisVAE reduces to the standard VAE, and when $G = K$, PDisVAE reduces to the fully disentangled VAE, e.g., $\beta$-TCVAE or FactorVAE.

**CelebaA.** The dataset contains 202,599 face images (Liu et al., 2015), cropped and rescaled to $(3, 64, 64)$. The encoder and decoder are deep CNN-based image-nets (Burgess et al., 2018). We fix the latent dimensionality $K = 12$ and vary the number of groups $G \in \{1, 2, 3, 4, 6, 12\}$. Training settings are similar to the previous experiments (see code for details).

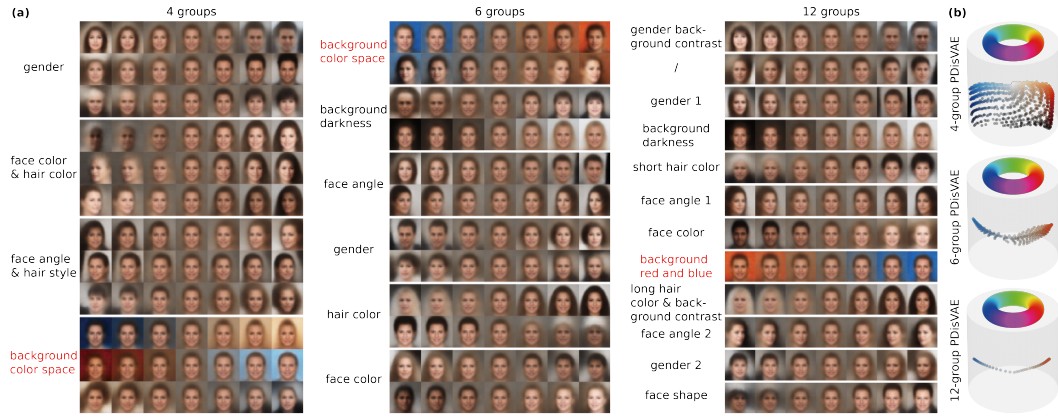

Figure 9: **(a)**: Reconstructed images are shown by varying one of the $K = 12$ latent dimensions from PDisVAE applied to the CelebA dataset, with different numbers of groups $G \in \{4, 6, 12\}$. Each row corresponds to varying one latent component (dimension) while fixing all others to 0s. **(b)** The spanned color space by the red-annotated color group in the $\{4, 6, 12\}$-group PDisVAE.

Fig. 9(a) shows the reconstructed images by varying each of the $K = 12$ components while fixing others as zero, for $G \in \{4, 6, 12\}$. The group meanings are annotated on the left. Particularly, with 4 or 6 groups, some attributes are represented by a group of higher rank rather than a single latent component, such as background color. Certain attributes are dependent on each other represented by a group, like the face color & hair color in the $G = 4$ setting. These important interpretations are harder to find by the fully disentangled $G = 12$ setting. Besides, fully disentangled VAE may fail to ensure perfect independence if the component setting and the true latent factor are largely mismatched (which is also hard to determine), like gender 1 and gender 2 in the $G = 12$ setting.

To understand how one semantic attribute is represented by multiple components within a group, we use background color as an example. The $G = 12$ groups setting in Fig. 9(a) shows that the background color is represented by a single component, which restricts the expression to a 1D color manifold as shown in $G = 12$ HSV cylinder in Fig. 9(b), which is not reasonable. With multiple latent components in a group representing background color, the background color can be expressed in 2D or 3D color manifolds as shown in $G = 6$ and $G = 4$ HSV cylinders, offering a more expressive and realistic representation. Results from all group settings are displayed in Fig. 14 in Appendix. A.4.

**Mouse dorsal cortex voltage imaging.** The dataset used in this study is a trial-averaged voltage imaging (method by Lu et al. (2023)) sequence from a mouse collected by us. It comprises 150 frames of $50 \times 50$ dorsal cortex voltage images, recorded while the mouse was subjected to a left-side air puff stimulus lasting 0.75 seconds. Each pixel is treated as a sample, and a linear model $x \sim \mathcal{N}(Az, \sigma^2 I)$ is learned. We investigate different numbers of groups $G \in \{1, 2, 3, 4, 6, 12\}$ while keeping the number of components constant at $K = 12$. Additionally, we explore fully

disentangled models by varying $K \in \{1, 2, 3, 4, 6, 12\}$ with $G = K$. The training procedures are similar to the previous experiments (see code for details).

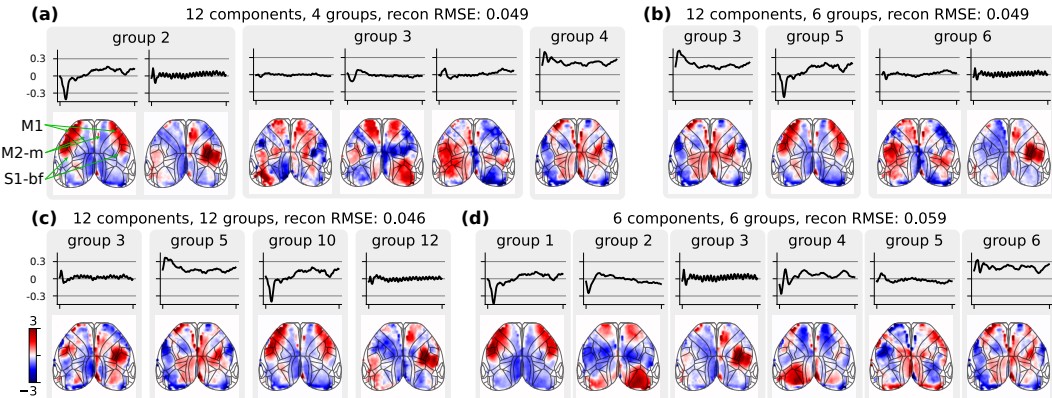

Figure 10: Brain maps $\left\{ \boldsymbol{z}_g^n \right\}_{n=1}^{50 \times 50}$ and the corresponding time series $\boldsymbol{A}_{:,g}$ from the learned groups by different PDisVAE configurations $(K, G)$, i.e., $K$ components, $G$ groups, and the group rank is $H = K/G$. Some groups contain dummy dimensions, so the effective group rank is lower than the specified group rank, and hence we only show those effective components.

Figure 10 shows the brain maps and corresponding time series learned from various PDisVAE configurations $(K, G)$. Learning $K = 12$ components with different $G$ groups (Fig. 10(a,b,c)) yields similar reconstruction RMSEs ($\approx 0.47$), but results in different latent representations. Assuming $G = 12$ as a fully disentangled model (Fig. 10(c)) is overly restrictive, as both group 3 and group 12 contain oscillations in the right primary somatosensory cortex-barrel field (S1-bf) and secondary motor cortex-medial (M2-m), demonstrating a lack of independence between these components. This configuration implies that there are not 12 independent components within this neural data. Conversely, assuming $G = 4$ groups (Fig. 10(a)) is insufficient, as group 2 mixes not only the oscillatory signals right S1-bf and M2-m but also signals from other regions like the right primary motor cortex (M1). This implies a failure to capture the complete scope of independence in the data. A $G = 6$ grouping (Fig. 10(b)) presents a more balanced approach. This model consists of six independent groups, each expressed by two latent components. Specifically, group 3's S1-bf and M2-m remain active, indicating these areas are stimulated during the air puff; group 6 is primarily responsible for the oscillations in S1-bf and M2-m, with minimal interference from the M1 signal. Moreover, the brain maps in group 2 from the 4-group configuration are effectively delineated into groups 5 and 6 in the 6-group configuration, further affirming the relative independence of M1 from S1-bf and M2-m during stimulus exposure.

The fully independent model with $(K, G) = (6, 6)$ (Fig. 10(d)) indicates that two components per group are necessary for accurate reconstruction. Specifically, having only one component per group is insufficient to reconstruct the raw video, as the RMSE for $(6, 6)$ is 0.059, which is significantly higher than the 0.049 RMSE for $(12, 6)$. The group reconstruction videos in the supplementary materials offer a more intuitive illustration of the full contribution of each group.

## 5 DISCUSSION

In this work, we propose the partially disentangled variational auto-encoder (PDisVAE) which is a more flexible method that can deal with group independence (partial disentanglement) in data, which is often a more realistic assumption than full independence (fully disentanglement) in a lot of applications. PDisVAE is a generalized method, which naturally reduces to standard VAE and fully disentangled VAE, by setting the number of groups to 1 or equal to the latent dimensionality. PDisVAE allows the existence of dummy latent components in groups if the number of learned latent components is less than the specified group rank. A potential limitation of PDisVAE is its need for an adequate number of groups and components to accurately express the disentangled latent space if the data requires, but we may not have guidance on this information. To address this, we might either try different configurations or develop techniques for group rank auto-reduction during training to enhance the performance, which could be a further direction.

## AUTHOR CONTRIBUTIONS

If you'd like to, you may include a section for author contributions as is done in many journals. This is optional and at the discretion of the authors.

## ACKNOWLEDGMENTS

Use unnumbered third level headings for the acknowledgments. All acknowledgments, including those to funding agencies, go at the end of the paper.

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

# A APPENDIX

## A.1 MARGINAL INDEPENDENCE

This part explains the sufficient but not necessary relationship between "group-wise independence" and "marginal independence". Consider latent variable $\boldsymbol{z} \in \mathbb{R}^M$ contains $M$ components that are independent between $G$ groups. The formal expression is

$$\overset{G}{\underset{g=1}{\perp}} \left( z_{(g-1)H+1}, \ldots, z_{gH} \right) \implies \bigwedge_{i \in g_1, j \in g_2, g_1 \neq g_2} z_i \perp z_j \tag{7}$$

but not vice versa. We start from the simple counterexample mentioned in Sec. 3.1 to explain why group-wise independence is a sufficient but not necessary condition of marginal independence.

Consider three random variables $z_1, z_2, z_3$ that follow the joint distribution shown in Tab. 2. Notice that $z_3$ is actually the exclusive or of the two others, i.e., $z_3 = \text{XOR}(z_1, z_2)$. It is obvious that $z_3 \not\perp (z_1, z_2)$ since when $z_1$ and $z_2$ are different, $p(z_3|z_1, z_2)$ is a discrete Dirac delta function at $z_3 = 0$; but when $z_1$ and $z_2$ are the same, $p(z_3|z_1, z_2)$ is a discrete Dirac delta function at $z_3 = 1$. Marginally, however, $z_1 \perp z_3$ and $z_2 \perp z_3$, since $p(z_3|z_1)$ is always a $p = 0.5$ Bernoulli distribution regardless of the value of $z_1$. The same arguments are also applicable to $z_2 \perp z_3$. Therefore, this counterexample shows that $z_1 \perp z_3, z_2 \perp z_3 \not\Longrightarrow (z_1, z_2) \perp z_3$. In other words, marginal independence does not imply group-wise independence.

Another way of checking this example is by the following theorem.

**Theorem 1.** $(x_1, \ldots, x_I) \perp (y_1, \ldots, y_J) \iff \left( f(x_1, \ldots, x_I) \perp g(y_1, \ldots, y_J) \, \forall \, \text{functions } f \text{ and } g \right).$

*Proof.* The $\implies$ is obvious. To prove $\impliedby$, simply taking $f$ and $g$ to be identity function, i.e., $f(x_1, \ldots, x_I) = (x_1, \ldots, x_I), g(y_1, \ldots, y_J) = (y_1, \ldots, y_J)$. $\square$

To check the example, consider the distribution of $(z_1 + z_2)$. $p(z_3|(z_1 + z_2) = 0)$ is a discrete Dirac delta function at $z_3 = 1$, which is different from $p(z_3|(z_1 + z_2) = 1)$ is a discrete Dirac delta function at $z_3 = 0$. Therefore, $(z_1, z_2) \not\perp z_3$.

To rigorously diagnose where $\impliedby$ breaks, we can write

$$p(z_1, z_2, z_3) = p(z_1|z_2, z_3)p(z_2, z_3) = p(z_1|z_2, z_3)p(z_2)p(z_3). \tag{8}$$

Note that in the last term, $p(z_1|z_2, z_3) \neq p(z_1|z_2)$. Specifically, $z_3$ cannot be removed just because of $z_1 \perp z_3$.

Table 2: The distribution table of $p(z_1, z_2, z_3)$.

| $z_1$ | $z_2$ | $z_3$ | $p(z_1, z_2, z_3)$ |
|-------|-------|-------|--------------------|
| 0 | 0 | 1 | 0.25 |
| 0 | 1 | 0 | 0.25 |
| 1 | 0 | 0 | 0.25 |
| 1 | 1 | 1 | 0.25 |

## A.2 DERIVATION OF PDISVAE

Given a dataset of $N$ equally treated samples, the probability of taking sample $n$ is $q(n) = \frac{1}{N}$, so that $\frac{1}{N} \sum_{n=1}^{N} [\cdot] = \mathbb{E}_{q(n)}[\cdot]$. Also let $q(\boldsymbol{z}|n) := q\left(\boldsymbol{z}|\boldsymbol{x}^{(n)}\right)$. When observing the whole dataset $\left\{\boldsymbol{x}^{(n)}\right\}_{n=1}^{N}$, the original target function to be maximized in VAE is

$$\text{ELBO}\left(\left\{\boldsymbol{x}^{(n)}\right\}_{n=1}^{N}; \theta, \phi\right) = \frac{1}{N} \sum_{n=1}^{N} \left[ \underbrace{\left(\mathbb{E}_{q(\boldsymbol{z}|\boldsymbol{x})}\left[\ln p\left(\boldsymbol{x}^{(n)}\big|\boldsymbol{z}\right)\right]\right)}_{\text{reconstruction log-likelihood}} - \underbrace{\text{KL}\left(q\left(\boldsymbol{z}\big|\boldsymbol{x}^{(n)}\right)\big\|p(\boldsymbol{z})\right)}_{\text{reverse KL divergence}} \right].$$
$$\tag{9}$$

$p(\boldsymbol{z})$ is a predefined prior distribution satisfying $p(\boldsymbol{z}) = \prod_{k=1}^{K} p(z_i)$, $\mathbb{E}[z_k] = 0$, $\mathrm{Var}[z_k] = 1$, $\forall m \in \{1, \ldots, M\}$, for example a standard normal (Gaussian) distribution. For simplicity, we also denote $\boldsymbol{z}_g = (z_{(g-1)H+1}, \ldots, z_{gH})$, so that $\boldsymbol{z} = (\boldsymbol{z}_1, \ldots, \boldsymbol{z}_G)$.

To derive the decomposed ELBO in PDisVAE (Eq. (5)), we focus on the aggregated reverse KL divergence:

$$
\frac{1}{N} \sum_{n=1}^{N} \mathrm{KL}(q(\boldsymbol{z}|n) \| p(\boldsymbol{z})) = \mathbb{E}_{q(n)} \left[ \mathrm{KL}(q(\boldsymbol{z}|n) \| p(\boldsymbol{z})) \right]
$$

$$
= \mathbb{E}_{q(n)} \mathbb{E}_{q(z|n)} \left[ \ln q(\boldsymbol{z}|n) - \ln p(\boldsymbol{z}) \right]
$$

$$
\text{code} = \mathbb{E}_{q(\boldsymbol{z},n)} \left[ \ln q(\boldsymbol{z}|n) - \ln q(\boldsymbol{z}) + \ln q(\boldsymbol{z}) - \ln \prod_{g=1}^{G} q(\boldsymbol{z}_g) + \ln \prod_{g=1}^{G} q(\boldsymbol{z}_g) - \ln \prod_{g=1}^{G} p(\boldsymbol{z}_g) \right]
$$

$$
= \mathbb{E}_{q(\boldsymbol{z},n)} \left[ \ln \frac{q(\boldsymbol{z}|n)p(n)}{q(\boldsymbol{z})p(n)} \right] + \mathbb{E}_{q(\boldsymbol{z})} \left[ \ln \frac{q(\boldsymbol{z})}{\prod_{g=1}^{G} q(\boldsymbol{z}_g)} \right] + \sum_{j=1}^{J} \mathbb{E}_{q(\boldsymbol{z}_g)} \left[ \ln \frac{q(\boldsymbol{z}_g)}{p(\boldsymbol{z}_g)} \right]
$$

$$
\text{math} = \underbrace{\mathrm{KL}(q(\boldsymbol{z},n) \| q(\boldsymbol{z})p(n))}_{\text{index-code mutual information}} + \underbrace{\mathrm{KL}\left( q(\boldsymbol{z}) \middle\| \prod_{g=1}^{G} q(\boldsymbol{z}_g) \right)}_{\text{partial correlation}} + \underbrace{\sum_{j=1}^{J} \mathrm{KL}(q(\boldsymbol{z}_g) \| p(\boldsymbol{z}_g))}_{\text{group-wise KL}},
$$

$$(10)$$

where $q(\boldsymbol{z}) = \sum_{n=1}^{N} q(\boldsymbol{z}, n) = \sum_{n=1}^{N} q\left(\boldsymbol{z}|\boldsymbol{x}^{(n)}\right) q(n)$ is the aggregated posterior, followed by Makhzani et al. (2015). In the derivation, we also used the theorem

$$
\mathbb{E}_{q(\boldsymbol{z},n)} f(\boldsymbol{z}) = \mathbb{E}_{q(n|\boldsymbol{z})q(\boldsymbol{z})} f(\boldsymbol{z})
$$

$$
= \int \sum_{n=1}^{N} q(n|\boldsymbol{z})q(\boldsymbol{z})f(\boldsymbol{z}) \, \mathrm{d}\boldsymbol{z}
$$

$$
= \int q(\boldsymbol{z})f(\boldsymbol{z}) \sum_{n=1}^{N} q(n|\boldsymbol{z}) \, \mathrm{d}\boldsymbol{z} \tag{11}
$$

$$
= \int q(\boldsymbol{z})f(\boldsymbol{z}) \, \mathrm{d}\boldsymbol{z}
$$

$$
= \mathbb{E}_{q(\boldsymbol{z})} f(\boldsymbol{z}),
$$

and similarly,

$$
\mathbb{E}_{q(\boldsymbol{z})} f(\boldsymbol{z}_g) = \mathbb{E}q(\boldsymbol{z}_{\backslash g}|\boldsymbol{z}_g)q(\boldsymbol{z}_g)f(\boldsymbol{z}_g) = \mathbb{E}_{q(\boldsymbol{z}_g)} f(\boldsymbol{z}_g). \tag{12}
$$

Note that it is clearer to use line 3 in Eq. (10) to implement the code.

### A.3 BATCH APPROXIMATION

#### A.3.1 IMPORTANCE SAMPLING

Although Eq. (6) in the main text intuitively gives the batch approximation, we still need a rigorous derivation to prove this is exactly the importance sampling (IS) we want. First, we have the aggregated posterior that can be expressed in different ways:

$$
q(z) = \sum_{n=1}^{N} q(z,n) = \sum_{n=1}^{N} q(z|n)q(n) = \frac{1}{N} \sum_{n=1}^{N} q(z|n) = \mathbb{E}_{q(n)}[q(z|n)]. \tag{13}
$$

However, to not confuse readers, we will keep the form $q(z) = \sum_{n=1}^{N} q(z,n)$ until the last step.

When we have a batch of size $M$: $\mathcal{B}_M \coloneqq \{n_1, n_2, \ldots, n_M\}$ (without replacement) and a particular sampled $z \sim q(z|n_*)$, where $n_* \in \mathcal{B}_M$, we want the importance sampling approximation of $q(z)$. According to Monte Carlo estimation,

$$
\hat{q}(z) = \frac{1}{M} \sum_{m=1}^{M} \frac{q(z, n_m)}{r(n_m)}, \tag{14}
$$

where $r$ is the proposal distribution. Note that $r(n_m) \neq \frac{1}{N}$, $\forall n_m \in \mathcal{B}$, since we must have $n_* \in \mathcal{B}_M$. Therefore, we need to understand the distribution of $r(n_m)$.

First, since we must have $n_* \in \mathcal{B}_M$, and the Monte Carlo estimation is the average on $\mathcal{B}_M$,

$$r(n_*) = \underbrace{1}_{n_* \text{ must be in } \mathcal{B}_M} \times \underbrace{\frac{1}{|\mathcal{B}_M|}}_{n_* \text{ is a Monte Carlo sample from } \mathcal{B}_M} = \frac{1}{M}. \tag{15}$$

Second, for other $n_m \notin \mathcal{B}_M$,

$$r(n_m) = \underbrace{\frac{\binom{N-2}{M-2}}{\binom{N-1}{M-1}}}_{n_m \text{ is selected in batch } \mathcal{B}_M} \times \underbrace{\frac{1}{|\mathcal{B}_M|}}_{n_m \text{ is a Monte Carlo sample from } \mathcal{B}_M} = \frac{M-1}{N-1} \frac{1}{M}. \tag{16}$$

$\binom{N-1}{M-1} = \frac{(N-1)!}{(M-1)!((N-1)-(M-1))!}$ is the number of all possible combinations of $\mathcal{B}_M$ that already contains $n_*$ (so we choose $M-1$ from the remaining $N-1$). $\binom{N-2}{M-2} = \frac{(N-2)!}{(M-2)!((N-2)-(M-2))!}$ is the number of all possible combinations of $\mathcal{B}_M$ that already contains $n_*$ and also contains $n_m$ (so we choose $M-2$ from the remaining $N-2$). Finally, we have

$$\begin{aligned}
\hat{q}(z) &= \frac{1}{M} \sum_{m=1}^{M} \frac{q(z, n_m)}{r(n_m)} \\
&= \frac{1}{M} \frac{q(z|n_*)q(n_*)}{r(n_*)} + \sum_{n_m \in (\mathcal{B}_M \backslash \{n_*\})} \frac{1}{M} \frac{q(z|n_m)q(n_m)}{r(n_m)} \\
&= \frac{1}{M} \frac{q(z|n_*)\frac{1}{N}}{\frac{1}{M}} + \sum_{n_m \in (\mathcal{B}_M \backslash \{n_*\})} \frac{1}{M} \frac{q(z|n_m)\frac{1}{N}}{\frac{M-1}{N-1}\frac{1}{M}} \\
&= \frac{1}{N}q(z|n_*) + \sum_{n_m \in (\mathcal{B}_M \backslash \{n_*\})} \frac{N-1}{M-1} \frac{1}{N} q(z|n_m).
\end{aligned} \tag{17}$$

### A.3.2 VARIANCE

From Chen et al. (2018), without loss of generality, assume $n_* = n_1$ and

$$\begin{aligned}
\text{MSS} &= \frac{1}{N}q(z|n_*) + \sum_{m=2}^{M-1} \frac{1}{M-1}q(z|n_m) + \frac{N-M+1}{N(M-1)}q(z|n_M) \\
&= \frac{1}{N}q(z|n_*) + \sum_{m=2}^{M-1} \frac{N}{M-1}\frac{1}{N}q(z|n_m) + \frac{N-M+1}{(M-1)}\frac{1}{N}q(z|n_M).
\end{aligned} \tag{18}$$

A sketch to compute the variances of the two methods is to think of them as sampled datasets of size $M$. Specifically, for IS, the inverse importance weights are a dataset of $\text{IS}_0 :=$
$$\left\{ 1, \underbrace{\frac{N-1}{M-1}, \ldots, \frac{N-1}{M-1}}_{M-1} \right\}.$$ For, MSS, the inverse importance weights are a dataset of $\text{MSS}_0 :=$
$$\left\{ 1, \underbrace{\frac{N}{M-1}, \ldots, \frac{N}{M-1}}_{M-2}, \frac{N-M+1}{M-1} \right\}.$$

There means are all $\frac{N}{M}$, since

$$\begin{cases} \overline{\text{MSS}_0} = \frac{1}{M}\left(1 + (M-2)\frac{N}{M-1} + \frac{N-M+1}{M-1}\right) = \frac{N}{M} \\ \overline{\text{IS}_0} = \frac{1}{M}\left(1 + (M-1)\frac{N-1}{M-1}\right) = \frac{N}{M} \end{cases} \tag{19}$$

Now we compute their variances.

$$\text{Var}[\text{MSS}] \propto \text{Var}[\text{MSS}_0]$$

$$= \frac{1}{M}\left[\left(1-\frac{N}{M}\right)^2 + (M-2)\left(\frac{N}{M-1}-\frac{N}{M}\right)^2 + \left(\frac{N-M+1}{M-1}-\frac{N}{M}\right)^2\right] \quad (20)$$

$$= \frac{2M^2 - (2N+2)M + N^2}{M^2(M-1)}.$$

$$\text{Var}[\text{IS}] \propto \text{Var}[\text{IS}_0]$$

$$= \frac{1}{M}\left[\left(1-\frac{N}{M}\right)^2 + (M-1)\left(\frac{N-1}{M-1}-\frac{N}{M}\right)^2\right] \quad (21)$$

$$= \frac{(N-M)^2}{M^2(M-1)}.$$

Since

$$\text{Var}[\text{IS}_0] - \text{Var}[\text{MSS}_0] = \frac{2-M}{M(M-1)} \leqslant 0, \ \forall M \geqslant 2, \quad (22)$$

the effectiveness of IS is higher, and hence IS is a more stable approximation than MSS.

### A.3.3 EMPIRICAL EVALUATION

To validate the aforementioned superiority of our proposed IS batch estimation method, we simulate a dataset consisting of 10 data points shown in Fig. 11(left). Each time, we run the three batch approximation methods on a batch of three randomly sampled points. We repeat this 1000 times and show their empirical evaluations in Fig. 11(right). Compared with the unbiased MWS estimator, MMS and IS are unbiased. Compared with MMS, the IS estimator has low empirical variance across 1000 repeats, which implies a more stable estimation.

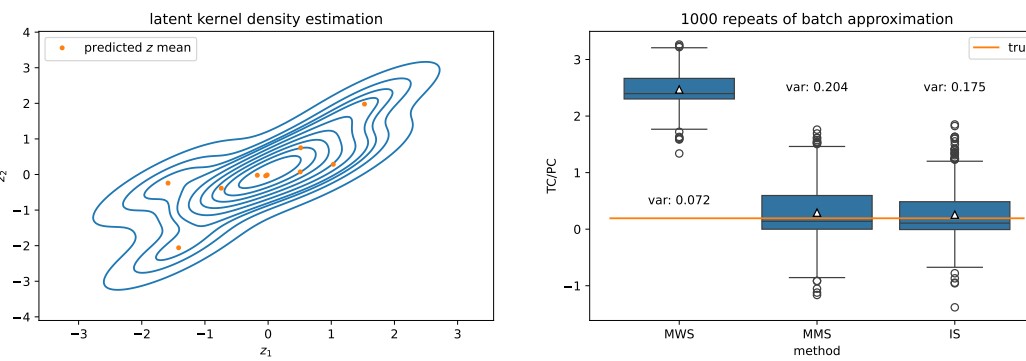

Figure 11: **Left**: Predicted mean of the latent $z = (z_1, z_2)$ and its kernel density estimation. **Right**: 1000 repeats of batch approximations by the three methods, their empirical variance across the 1000 repeats.

### A.4 SUPPLEMENTARY RESULTS

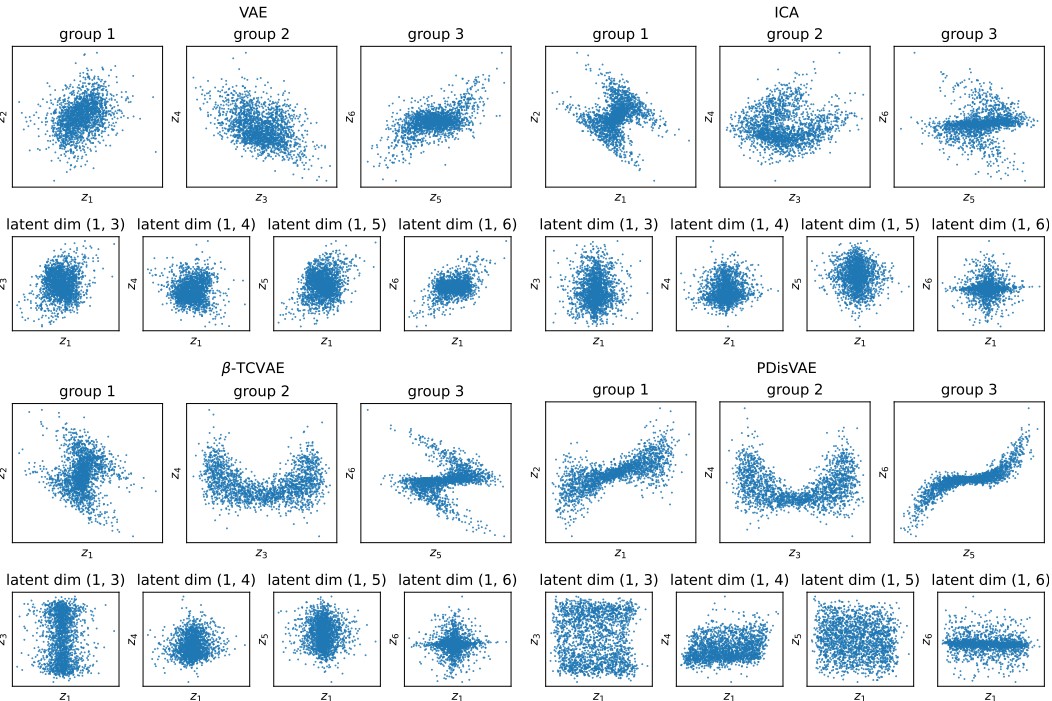

Figure 12: Latent alignment results of different methods. Each group is aligned and matched to the true latent shown in Fig. 2(a). Some between-group pairs are also plotted to visually understand the marginally independent distributions between groups.

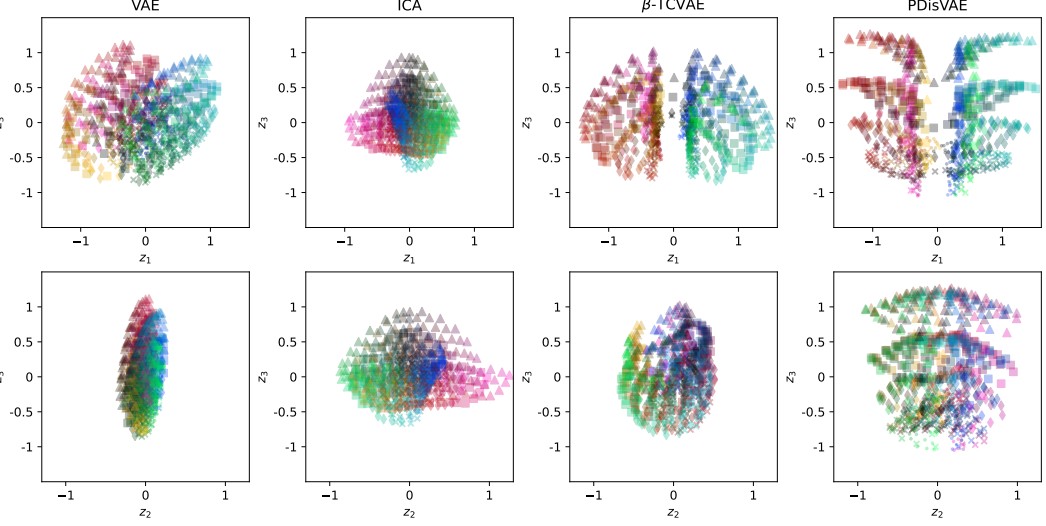

Figure 13: The latent plot after alignment in latent space $(z_1, z_3)$ and $(z_2, z_3)$ for different methods. The color representation for location is the same as the color representation in Fig. 7(a), and the marker of the point in the latent plots represents the size of the square in the observation images.

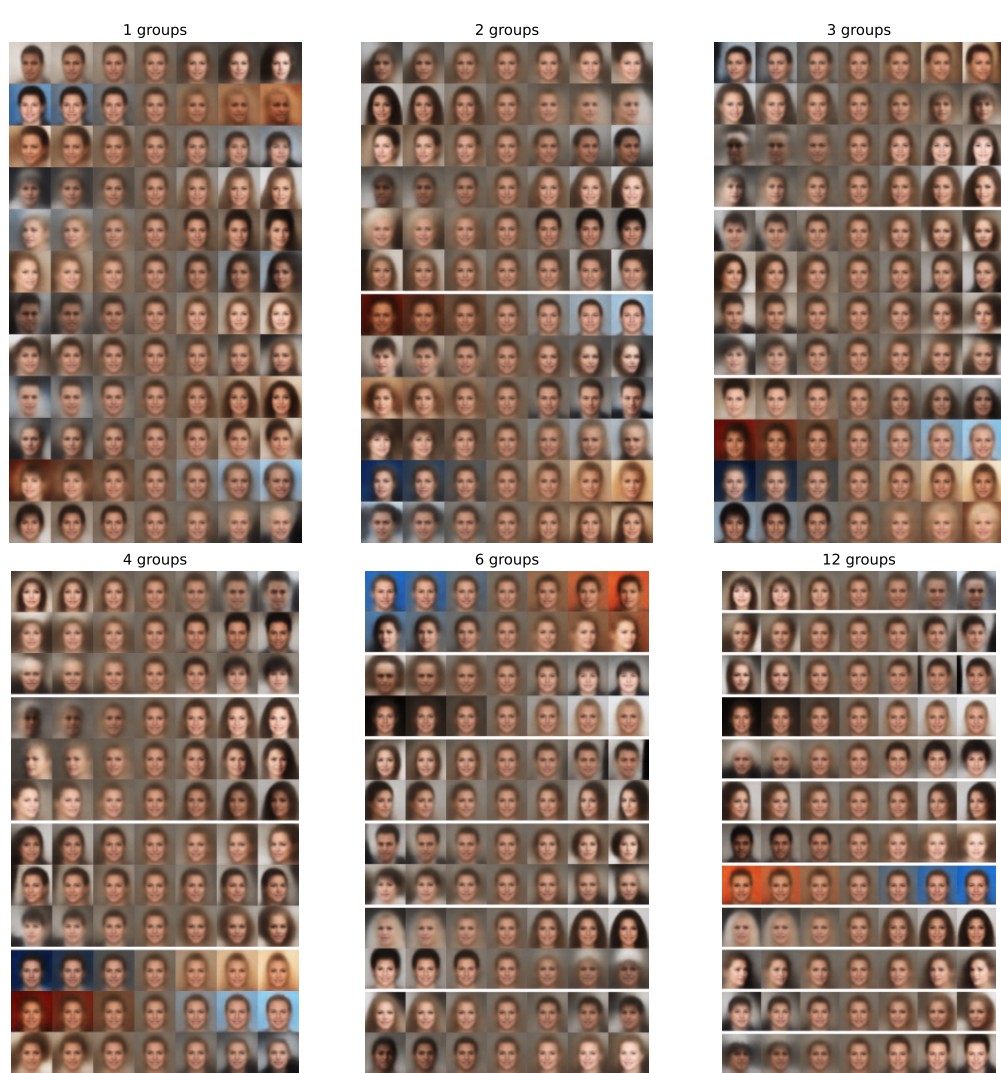

Figure 14: The reconstructed images by varying one of the $K = 12$ disentangled latent from applying PDisVAE to the CelebA dataset with the different number of groups $G \in \{1, 2, 3, 4, 6, 12\}$. When $G = 1$, PDisVAE becomes the standard VAE; when $G = K = 12$, PDisVAE becomes the fully entangled VAE (e.g., $\beta$-TCVAE or FactorVAE). In each plot, each row is by varying one latent component (latent dimension) while fixing all others to 0s.

## A.5 Related works

Realizing there are a lot of methods related to latent disentanglement, we provide Tab. 3 with a list to summarize their contributions and differences.

Table 3: Related works

|  | full disentanglement | partial disentanglement |
| --- | --- | --- |
| By prior (not flexible) | [1] | [4] |
| By extra penalty to loss (flexible) | [2][3] | Our PDisVAE |
| By auxiliary information (supervised) | [7] |  |
| Others | [5][6][8][9][10] |  |

- [1] ICA (Hyvärinen & Oja, 2000): Traditional ICA uses a non-Gaussian prior to achieving full disentanglement since independence is non-Gaussian from the statistical perspective. However, the choice of the non-Gaussian prior is critical and might be too rigid, hurting the flexibility of the method.
- [2] FactorVAE (Kim & Mnih, 2018) [3] $\beta$-TCVAE (Chen et al., 2018): These two papers start from the statistical definition of full independence to add an extra total correlation to achieve full independence rigorously. The only difference between these two papers is their implementations of minimizing TC.
- [4] ISA-VAE (Stühmer et al., 2020): ISA-VAE realized the commonly existing group-wise independence (partial disentanglement) in the real-world data. It utilizes a group-wise independent prior called $L^p$-nested distribution to achieve the partial disentanglement. However, they did not validate their approach on partially disentangled synthetic datasets, but merely evaluated their approach using fully disentangled assumptions for dsprites and CelebA datasets.
- [5] $\beta$-VAE (Burgess et al., 2018): Directly penalize the KL divergence of the VAE ELBO loss, in which TC (in Eq. (10)) is implicitly penalized. This approach has been proven to be worse than $\beta$-VAE and FactorVAE.
- [6] (Locatello et al., 2019): This research presented common challenges in finding disentangled latent through an unsupervised approach, implying supervision with semantic latent labels might be necessary under the assumption of full latent disentanglement. This also gives us a hint that full disentanglement might be a strong and inappropriate assumption and could result in poor latent interpretation.
- [7] (Ahuja et al., 2022): This paper uses weak supervision from observations generated by sparse perturbations of the latent variables, which requires auxiliary information to the latent variables.
- [8] (Meo et al., 2024): This paper replace the traditional TC term with a novel TC lower bound to achieve not only disentanglement but generalized observation diversity.
- [9] (Bhowal et al.): This paper claims that VAE with orthogonal structure could also achieve latent full disentanglement.
- [10] (Hsu et al., 2024): The full disentanglement is achieved by a technique called latent quantization. The approach is quantizing the latent space into discrete code vectors with a separate learnable scalar codebook per dimension. Besides, weight decay is also applied to the model regularization for better full disentanglement.

## A.6 MORE RESULTS ON THE PDSPRITES DATASET

Tab. 4 presents the partial correlation (PC), latent $R^2$, MSE, and mutual information gap (MIG) evaluated for the partial disentanglement (group-wise independence) in the estimated latent.

Since only ISA (Stühmer et al., 2020) and our PDisVAE explicitly require group-wise independence rather than strong full independence, only ISA and our PDisVAE obtain the lowest PC. Compared with ISA, PDisVAE is more flexible since we achieve partial disentanglement by adding a PC penalty term to the loss function. Via the PC penalty, the latent distribution is estimated through the aggregated posterior $q(\boldsymbol{z})$ from the learned decoder, rather than a fixed $L^p$-nested prior in ISA. Therefore, PDisVAE obtains more accurate and partially disentangled latent when evaluated with the true latent (labels).

Table 4: Different metrics evaluated on the pdsprites dataset.

|  | PC $\downarrow$ | $R^2 \uparrow$ | MSE $\downarrow$ | MIG $\uparrow$ |
|---|---|---|---|---|
| VAE | 1.01 (0.02) | 0.22 (0.04) | 0.29 (0.02) | 0.15 (0.10) |
| ICA | 1.76 (0.07) | 0.22 (0.06) | 0.28 (0.03) | 0.14 (0.09) |
| ISA | **0.70 (0.01)** | 0.23 (0.02) | 0.33 (0.01) | 0.24 (0.08) |
| $\beta$-TCVAE | 0.91 (0.10) | 0.33 (0.06) | **0.24 (0.04)** | 0.36 (0.13) |
| $\alpha$-TCVAE | 1.84 (0.03) | 0.31 (0.02) | 0.27 (0.01) | 0.29 (0.09) |
| PDisVAE | **0.68 (0.04)** | **0.54 (0.08)** | **0.23 (0.04)** | **0.49 (0.07)** |

