# OpenReview forum: "A Revisit of Total Correlation in Disentangled Variational Auto-Encoder with Partial Disentanglement"
_ICLR.cc/2025/Conference — Submitted to ICLR 2025_

### Official Review · Reviewer_93Kr · 2024-10-29

**Soundness:** 2
**Presentation:** 3
**Contribution:** 3
**Rating:** 6
**Confidence:** 4

**Summary:**

The partially disentangled VAE is proposed to enforce groups of latent dimensions independent of each other. Within the group, the latent variables are correlated. The method outperforms baselines ($\beta$-VAE and $\beta$-TCVAE) in synthetic datasets, providing interpretable features for real datasets.

**Strengths:**

The paper's topic is relevant, and the idea behind the method is valuable.

**Weaknesses:**

While I appreciate the interest behind the partial correlation term in (5), I am not convinced about the paper in many aspects.

1) First of all, the writing style is sometimes vague and there and imprecise:

- "The core idea inspired by ICA is that non-Gaussian is independent", This statement seems oversimplified.

- " if the true number of disentangled latent components is two but we instruct the logcosh-priored VAE to find three, it will yield three components with poor disentanglement instead of finding two disentangled components and one non-informative component" Is there empirical evidence or theoretical justification for this claim about logcosh-priored VAE behavior? If so, could you provide it?

2) Important recent literature is overlooked. After a quick search I found:

- $\alpha$TC-VAE: On the relationship between Disentanglement and Diversity (ICLR 2024)
- Why do Variational Autoencoders Really Promote Disentanglement? (ICML 2024)
- Disentanglement via Latent Quantization (NeurIPS 2023)

    How do the methods in these papers compare to the proposed method in terms of handling group-wise independence?
    Are there specific metrics or experimental setups from these papers that would be particularly informative to include in the comparison?
    Do any of these papers address partial disentanglement, and if so, how does their approach differ from the one proposed here?

3) I cannot assess how conclusive the experimental results are over real data sets.

-  Provide CelebA results for $\beta$-TCVAE and FactorVAE, or explain why these were not included.
- Describe their hyperparameter tuning process for all methods, including baselines. What range of values were explored for each hyperparameter?
- Report the best hyperparameter configurations found for each method, along with the performance achieved.
- If possible, include a sensitivity analysis showing how performance varies with key hyperparameters for each method.

4) Scalability and sensitivity. It would be important to analyze at least the following points:;

- Testing the method with increasing latent dimensions (e.g., 32, 64, 128) and reporting performance trends.
- Analyzing the impact of different grouping strategies on performance for a fixed latent dimension. For example, comparing random groupings vs. learned groupings.
- Evaluating the method's performance with increasingly complex decoder architectures, and reporting how this affects both disentanglement and reconstruction quality.
- Discussing potential strategies for determining optimal groupings in high-dimensional spaces, or acknowledging this as a limitation if no clear solution exists.


Minor typo: q(z), the aggregated posterior, is not defined in (1)

**Questions:**

See above

---

> ### Author Response · Authors · 2024-11-20
>
> Dear Reviewer 93Kr,
>
> Thank you very much for your time and valuable comments on our paper. We highly appreciate your recognition of the strengths of our paper. Hopefully, the following responses could resolve most of your concerns and answer your questions.
>
> > 1.1 "The core idea inspired by ICA is that non-Gaussian is independent", This statement seems oversimplified.
>
> Please kindly notice that, this sentence is not originally stated by us, but cited from the [FastICA paper by Hyvarinen and Oja](https://www.sciencedirect.com/science/article/pii/S0893608000000265). This sentence is exactly the title of their Sec. 4.1, which illustrates the reason. The intuition is also illustrated in Fig. 7, and the FastICA algorithm is based on this idea, which is explained in Sec. 4 of their paper.
>
> > 1.2 "if the true number of disentangled latent components is two but we instruct the logcosh-priored VAE to find three, it will yield three components with poor disentanglement instead of finding two disentangled components and one non-informative component" Is there empirical evidence or theoretical justification for this claim about logcosh-priored VAE behavior? If so, could you provide it?
>
> Please kindly notice that the empirical result showing this issue was already in Fig. 6 of our initial submission. Specifically, there are three fully independent latent components but ICA is asked to find six. The orange curves show the poor latent estimated by ICA. However, the green and red curves show that $\beta$-TCAVE and our PDisVAE is able to find three fully independent latent components and three dummy latent components, implying that penalizing TC/PC is a more flexible way for latent disentanglement than assuming a non-Gaussian prior (ICA). Although there is no theoretical proof on this point, we do often observe ICA finish with "pseudo" independent components, which are not really independent after a detailed checking.
>
> > 2 Important recent literature is overlooked. After a quick search I found:
>
> Thanks for reminding us of these important papers. We apologize that there should have been a section on related works. To answer similar questions asked by other reviewers, we have added a section of related works in Appendix A.5 as an early response.
>
> > 3.1 Provide CelebA results for $\beta$-TCVAE and FactorVAE, or explain why these were not included.
>
> Please kindly notice that the empirical results showing this issue were already in Fig. 9 and Fig. 13 in our initial submission. Sec. 4.4, paragraph 1, the last sentence, claims that "when $G=K$, PDisVAE reduces to the fully disentangled VAE, e.g., $\beta$-TCVAE or FactorVAE". So, the results are in the $G=12$ groups setting in Fig. 9 and Fig. 13. This is in fact an important point that our PDisVAE is flexible enough to generalize to standard VAE and fully disentangled VAE with particular group configurations.
>
> > 3.2-3.4 regarding hyperparameters.
>
> In real-world datasets, we don't know the true number of independent groups and each group's rank. So, we tried a series of group configurations. Different group configurations have different disentanglement interpretations as shown in Fig. 9. It should not be assumed as a hyperparameter, but a flexibility of the method. With higher group rank, the latent will be more expressive but the disentanglement will be weaker. The hyperparameter of PDisVAE is $\beta$, which was already discussed in Fig. 4 of our initial submission. The optimal choice of $\beta$ is similar to the findings in $\beta$-TCVAE, which is $\beta\in(2, 10)$, and for a fair comparison, we choose $\beta=4$ for our PDisVAE and $\beta$-TCVAE.
>
> > 4.1 Testing the method with increasing latent dimensions (e.g., 32, 64, 128) and reporting performance trends.
> > 4.2 Analyzing the impact of different grouping strategies on performance for a fixed latent dimension. For example, comparing random groupings vs. learned groupings.
>
> Please kindly notice that in real-world experiments of Sec. 4.4, we did exactly this ablation study. The results are reported in Fig. 9, Fig. 10 and the group reconstruction videos were in the supplementary material of our initial submission.

---

> > ### Author Response · Authors · 2024-11-20
> >
> > > 4.3 Evaluating the method's performance with increasingly complex decoder architectures, and reporting how this affects both disentanglement and reconstruction quality.
> >
> > We acknowledge that encoder and decoder architectures influence the disentanglement and reconstruction quality. The strategy is that, in synthetic 1 and synthetic 2, we know the true generating process, so we choose the encoder and decoder to comply with the generating process. For pdsprites and CelebA, we don't know the true generating process, so we pick the most commonly used encoder-decoder architecture from those famous disentanglement papers like [$\beta$-VAE](https://arxiv.org/abs/1804.03599) and [$\beta$-TCVAE](https://proceedings.neurips.cc/paper/2018/hash/1ee3dfcd8a0645a25a35977997223d22-Abstract.html), for a consistent and fair comparison.
> >
> > However, we kindly think that the choice of the VAE architecture is out of the contribution scope of our PDisVAE and the $\beta$-TCVAE/factor-VAE papers. For all these VAE-based disentanglement methods, a more powerful VAE architecture could be more likely to achieve better reconstruction and latent disentanglement. Investigating this aspect could be a big open question to the field of representational learning and the VAE-based disentanglement method, which is out of our paper's aim of contribution.
> >
> > > 4.4 Discussing potential strategies for determining optimal groupings in high-dimensional spaces, or acknowledging this as a limitation if no clear solution exists.
> >
> > Please kindly notice that this point was already discussed in Sec. 5, sentences -2 and -1, of our original submission.
> >
> > > Minor typo: $q(z)$.
> >
> > Thanks for this catch. We have moved the aggregated posterior definition from Sec. 3.1 to Sec. 2.

---

> > > ### Comment · Reviewer_93Kr · 2024-11-22
> > >
> > > Thank you very much for your responses. I realize now that some of my questions were already addressed in the original submission, and I apologize for the oversight.
> > >
> > > While I am inclined to raise my score, I still believe that including a related work section alone is insufficient. At the very least, a direct experimental comparison with the $\alpha$TC-VAE method is essential, given its strong relevance to your approach.

---

> ### Author Response · Authors · 2024-11-22
>
> Thank you for your recognition of our revision and the suggestion. To answer similar comparison questions asked by other reviewers, we have added a comprehensive table comparison in Appendix A.6 to show the superiority of our PDisVAE. Some metrics and interpretations were already presented in Fig. 7 and Fig. 8 of our initial submission. For your convenience, we show the comparison results here:
>
> The table below presents the partial correlation (PC), latent $R^2$, MSE, and mutual information gap (MIG) evaluated for the partial disentanglement (group-wise independence) in the estimated latent.
>
> |                | PC $\downarrow$      | $R^2$ $\uparrow$     | MSE $\downarrow$     | MIG $\uparrow$       |
> |----------------|----------------------|----------------------|----------------------|----------------------|
> | VAE            | 1.01 (0.02)          | 0.22 (0.04)          | 0.29 (0.02)          | 0.15 (0.10)          |
> | ICA            | 1.76 (0.07)          | 0.22 (0.06)          | 0.28 (0.03)          | 0.14 (0.09)          |
> | ISA            | **0.70 (0.01)** | 0.23 (0.02)          | 0.33 (0.01)          | 0.24 (0.08)          |
> | $\beta$-TCVAE  | 0.91 (0.10)          | 0.33 (0.06)          | **0.24 (0.04)** | 0.36 (0.13)          |
> | $\alpha$-TCVAE | 1.84 (0.03)          | 0.31 (0.02)          | 0.27 (0.01)          | 0.29 (0.09)          |
> | PDisVAE        | **0.68 (0.04)** | **0.54 (0.08)** | **0.23 (0.04)** | **0.49 (0.07)** |
>
> ISA and $\alpha$-TCVAE are the two new alternative methods. Since only ISA and our PDisVAE explicitly require group-wise independence rather than strong full independence, only ISA and our PDisVAE obtain the lowest PC. Compared with ISA, PDisVAE is more flexible since we achieve partial disentanglement by adding a PC penalty term to the loss function. Via the PC penalty, the latent distribution is estimated through the aggregated posterior $q(\boldsymbol z)$ from the learned decoder, rather than a fixed $L^p$-nested prior in ISA. For $\alpha$-TCVAE, it is classified as a fully disentangled method but our pdsprites dataset contains group-wise independent latent, therefore, PDisVAE obtains more accurate and partially disentangled latent when evaluated with the true latent (labels).

---

> > ### Comment · Reviewer_93Kr · 2024-11-25
> >
> > Thanks for your reply and for including the new methods in the experimental evaluation. I believe the paper has really improved since the original submission, and I will update the score accordingly.

---

> > > ### Author Response · Authors · 2024-11-25
> > >
> > > Thank you very much for your recognition of our improvement!

---

### Official Review · Reviewer_1ysP · 2024-11-02

**Soundness:** 2
**Presentation:** 1
**Contribution:** 2
**Rating:** 1
**Confidence:** 5

**Summary:**

The paper introduces the Partially Disentangled VAE (PDisVAE) as a flexible extension to fully disentangled VAEs, addressing the issue of over-restrictive full independence assumptions in latent representations. By introducing partial correlation (PC) instead of total correlation (TC), the model can achieve group-wise disentanglement, which can be more suitable for complex data.  The paper provides a generalized formulation of partial correlation for group-wise independence and proposes an optimal importance sampling (IS) batch approximation to reduce the high variance issue seen in traditional TC computation methods

**Strengths:**

- The authors introduce the Partially Disentangled VAE (PDisVAE), which effectively addresses the limitations of fully disentangled VAEs by permitting group-wise disentanglement. This focus aligns well with current literature trends in the field of disentangled VAEs.

- The paper highlights the high variance issues present in existing batch approximation methods for total correlation (TC) and proposes an optimal importance sampling (IS) batch approximation for partial correlation (PC) as a solution to this challenge.

- Experiments conducted on synthetic datasets demonstrate the model's capability to effectively manage group-wise independence.

**Weaknesses:**

**Structure of the Paper**

- The current structure could be refined for clarity. The introduction, in particular, could be reworked to clearly convey why disentanglement is needed and how incorporating total correlation (TC) or partial disentanglement can address specific limitations. This would help readers follow the paper’s motivation more naturally.
- Additionally, the use of `$ \vspace $` throughout the document, for example in lines 24–26, 197–200, and 370–371, does not align with ICLR formatting guidelines. Frequent use of `$ \vspace $` reduces readability, creating dense blocks of text. To improve flow, consider adjusting content and shortening details in the experimental section where information is clear from figures.

**Notation and Clarity Issues**

- There are a few instances where the notation could be made clearer. For example, `$ p(z|n_{*}) $` is introduced without a clear distinction from `$ p(z|n) $`, and `$ N $` in the summation notation is somewhat ambiguous. If `$ n_{*} $` refers to a specific data point, this should be clarified. Additionally, in line 206, batch notation could be simplified for readability by avoiding complex subscripts or just referring to "batch" directly.

**Context of Literature and Definition of Disentanglement**

- The paper introduces claims related to disentanglement but does not define the concept in enough detail. It would strengthen the paper to clarify at the beginning why disentanglement is challenging to achieve directly with VAEs.
- The authors might reference prior results indicating these challenges (which is not the case in the current version), such as those found in [Locatello et al., 2019] and [Ahuja et al., 2022]

**Clarification on Importance Sampling (IS) Approach**

- The authors state that their importance sampling (IS) approach in Table 1 improves prior methods, yet the differences and underlying assumptions remain unclear. It would be helpful to explicitly describe how this IS approach differs from previous work and to specify any variance assumptions. This would help readers better understand the novelty and practical impact of the method.

**Metrics for Assessing Disentanglement**

- The metrics selected, particularly $ R^2 $, assess identifiability but may not fully capture disentanglement. Complementing $ R^2 $ with additional metrics like DCI or RMIG (see [Eastwood et al., 2022] and [Carbonneau et al., 2022]) would provide a more comprehensive view of group-wise disentanglement. This addition would enhance the evaluation and provide stronger evidence of the model’s effectiveness.

**Concerns with Theorem 1 and Its Proof**

- In Theorem 1, the statement that $ (x_1, \ldots, x_I) \perp (y_1, \ldots, y_J) \iff f(x_1, \ldots, x_I) \perp g(y_1, \ldots, y_J) \ \forall $ functions $ f $ and $ g $ is not fully substantiated. While the forward implication $ \Rightarrow $ holds, the backward implication $ \Leftarrow $ requires additional assumptions.
- The authors attempt to establish the $ \Leftarrow $ direction by setting $ f $ and $ g $ as identity functions. However, this choice does not sufficiently demonstrate general independence of $ (x_1, \ldots, x_I) $ and $ (y_1, \ldots, y_J) $, as the independence of all functions of $ x $ and $ y $ does not imply the independence of $ x $ and $ y $ themselves. A counterexample, such as when $ x $ and $ y $ are jointly normal with non-zero correlation, illustrates this point.
- This portion would benefit from either additional assumptions to support the implication or a reformulation of the theorem to avoid overstating the result.

**References:**
- [1] Locatello, Francesco, et al. "Challenging common assumptions in the unsupervised learning of disentangled representations." International Conference on Machine Learning. PMLR, 2019.
- [2] Ahuja, K., Hartford, J. S., & Bengio, Y. "Weakly supervised representation learning with sparse perturbations." Advances in Neural Information Processing Systems, 35, 2022.
- [3] Eastwood, Cian, et al. "DCI-ES: An extended disentanglement framework with connections to identifiability." arXiv preprint arXiv:2210.00364, 2022.
- [4] Carbonneau, Marc-André, et al. "Measuring disentanglement: A review of metrics." IEEE Transactions on Neural Networks and Learning Systems, 2022.

**Questions:**

1. **Use of $R^2$ for Assessing Disentanglement**: The paper utilizes $R^2$ as a metric to evaluate disentanglement. What justification do the authors provide for this choice, and how is $R^2$ theoretically connected to the concept of disentanglement? There appears to be ambiguity regarding how $R^2$ specifically relates to the disentanglement characteristics of the groups. For more information on alternative metrics, please refer to the weaknesses section.

2. **Comparison with Other Models**: How does the partial correlation term in PDisVAE theoretically improve the learning of latent representations when compared to existing methods like [5], including recent works that utilize Hausdorff distances to achieve independence of support instead of strict independence [6]? Are there particular theoretical guarantees or properties that PDisVAE upholds?

3. **Impact of Group Size**: In your experiments, varying group ranks may present challenges. From a theoretical perspective, how does the size of the groups influence PDisVAE’s ability to recover meaningful latent structures? Are there specific thresholds or bounds that could affect its performance?

4. **Variance in Estimates**: The authors highlight the high variance present in batch approximations for total correlation (TC). What theoretical considerations contributed to the creation of the optimal importance sampling method, and in what ways does it address these variance-related issues?

**References:**

[5] Yao, Dingling, et al. "Multi-view causal representation learning with partial observability." arXiv preprint arXiv:2311.04056 (2023).

[6] Roth, Karsten, et al. "Disentanglement of correlated factors via hausdorff factorized support." arXiv preprint arXiv:2210.07347 (2022).

---

> ### Author Response · Authors · 2024-11-21
>
> Dear Reviewer 1ysP,
>
> Thank you very much for your time and valuable comments on our paper. We highly appreciate your recognition of the strengths of our paper. Hopefully, the following responses could resolve most of your concerns and answer your questions.
>
> > Structure of the Paper
>
> We apologize for the clarity and the use of `vspace`. We will reorganize our draft once we finish it and send you a notification.
>
> > Notation and Clarity Issues
>
> Please kindly notice that
> * $N$ is the number of data points (samples) was already introduced in the first sentence of Sec. 2.1 of our initial submission "Given a dataset of observations $\\{x^{(n)}\\}_{n=1}^N$ consisting of $N$ samples..."
> * $n_*$ refers to a specific data point was already clarified in line 206 (Sec. 2.4, paragraph 2, sentence 1) of our initial submission "Intuitively, when we only have a batch $\mathcal B_M \subsetneqq \\{1,\dots,N\\}$ and a sampled $z\sim q(z|n_*)$, where $n_*$ is a specific example point in $\mathcal B_M$..."
> * We kindly think that the batch notation here is necessary for solid and rigorous math clarification.
>
> > Context of Literature and Definition of Disentanglement and Question 2
>
> * Thanks for this point. We would like to claim at the beginning of the paper that disentanglement cannot be plainly achieved by naive VAE.
> * We have added a section of related works to Appendix A.5. We will reorganize our draft once we finish it and send you a notification.
> * As stated in texts below Eq. 5, Eq. 2 of [$\beta$-TCVAE](https://proceedings.neurips.cc/paper/2018/hash/1ee3dfcd8a0645a25a35977997223d22-Abstract.html), and [Wikipedia](https://en.wikipedia.org/wiki/Total_correlation), total correlation (TC) is a natural way of requiring independence from the statistical definition of independence. Our partial correlation (PC) is merely a natural generalization from full independence to group-wise independence. In other words, minimizing PC is an intuitive and explicit way of reducing independence between latent groups from the statistical definition of independence, i.e., Eq. 4 of our initial submission.
> * To answer similar comparison questions asked by other reviewers, we have added a comprehensive table comparison in Appendix A.6.
>
> > Clarification on Importance Sampling (IS) Approach and Question 4
>
> Thanks for this point. Please kindly notice that the rigorous derivation of the claims in Tab. 1 was already linked to Appendix 3 in our initial submission. To show the superiority of our IS batch approximation method, comparison results have been added to Appendix 3.3. The intuition behind this contribution is that we need a good estimation of TC/PC terms to achieve independence. Without a stable and accurate estimation of TC/PC, it would be harder to achieve the desired independence penalized by TC/PC.
>
> > Metrics for Assessing Disentanglement and Question 1
>
> Thanks for this question. For synthetic datasets, we know the true disentangled (independent) latents (labels), so we compute the $R^2$ metric between the estimated latents and the true as a supervised measurement. The pairwise $t$-test of not only the supervised latent $R^2$ but also the unsupervised partial correlation (PC) in Fig. 2(b), and Fig. 5(b) rigorously show significantly better results from our PDisVAE.
>
> Since MIG is a commonly used metric for full disentanglement, we have tried to adapt it to evaluate partial disentanglement (group-wise independence) and presented a table including both $R^2$ and MIG in Appendix A.6.
>
> > Concerns with Theorem 1 and Its Proof
>
> Thanks for this point. We realized that we should add a parenthesis in Theorem 1 to eliminate ambiguity, i.e.,
>
> $(x_1, \dots, x_I) \perp (y_1, \dots, y_J) \iff \big(f(x_1,\dots, x_I) \perp g(y_1, \dots, y_J)$ $\forall$ functions $f$ and $g\big)$
>
> The $\impliedby$ is stated as for all functions $f$ and $g$, i.e., if arbitrary choice of $f$ and $g$ makes $f(\boldsymbol x) \perp g(\boldsymbol y)$, then $\boldsymbol x\perp \boldsymbol y$. Since arbitrary choice of $f$ and $g$ holds, then $f = g = \operatorname{identity}$ necessarily holds, i.e., $\boldsymbol x\perp \boldsymbol y$.
>
> > Question 3 about group size
>
> The setting of group size is an open question similar to the setting of latent dimensionality.
> * Synthetic dataset 2 shows that PDisVAE can find a dummy latent if the true group rank is less than the specified group rank.
> * With larger latent dimensionality, we are less likely to omit existed independent group.
>
> Therefore, in the CelebA dataset, for example, we show that having a group rank greater than 1 (group independence) rather than group rank equal to 1 (full independence) can provide more interpretable disentanglement, especially Fig. 9(b). This just indicates the significance of partial disentanglement (group rank higher than 1).
>
> Similar to the full disentanglement models like $\beta$-TCVAE and even all VAEs, it is meaningless to have a huge latent space and this can also hurt the learning effects.

---

> > ### Author Response · Authors · 2024-11-27
> > **Sincerely looking forward to your feedback before deadline**
> >
> > Dear Reviewer 1ysP,
> >
> > Thank you very much again for your initial review of our paper. Given it is close to the discussion deadline, and we are not sure whether you are satisfied with our answers and revised paper or whether there are still some concerns that have not been fully resolved, we would like to kindly remind you that we have answered all your questions and concerns mentioned in the weaknesses. Our latest revision and new materials in the appendix provide a clearer picture of our work. We appreciate your time and invaluable feedback.
> >
> > Best,

---

> > ### Comment · Reviewer_1ysP · 2024-11-30
> > **Reservations regarding Theorem 1 and its proof**
> >
> > Dear authors,
> >
> > Thank you for addressing my concerns. However, I still have significant reservations regarding **Theorem 1** and its proof.
> >
> > - Regarding Theorem 1, the proof claims that we can choose $ f $ and $ g $ as identity functions, which makes the backward implication trivial. However:
> >   1. The independence of all $ f(x\_1, \ldots, x\_I) $ and $ g(y\_1, \ldots, y\_J) $ does not guarantee that the joint distribution of $ (x\_1, \ldots, x\_I) $ and $ (y\_1, \ldots, y\_J) $ factorizes.
> >   2. The independence of identity functions $ f $ and $ g $ only tests the original random variables directly, not the transformed ones.
> >
> > - While I appreciate your response, I remain doubtful that the authors fully understand the disentanglement mechanism and how it can be properly measured using $ R^2 $. Adding MIG does not effectively assess the concept of *group independence* proposed in the paper. To the best of my knowledge, the limitation of MIG lies in the fact that it only verifies whether the information related to a factor is captured by a single code dimension (the paper proposes more this type of disentanglement "group-wise independence"). However, it does not directly measure modularity that your looking for in that paper. For example, a single code dimension might still encode information about multiple factors/group factors. JEMMIG (K. Do, 2020) addresses this limitation by incorporating the joint entropy of the factor and its best code.
> >
> > - I also tried to understand your derivation of the ELBO in Appendix 1.3. Could you clarify what you mean by "code" in line 713 and "math" in line 719?
> >
> > Based on these points, I still do not see that the paper presents a clear contribution and needs some reformulation.
> >
> > **References**
> > [1] K. Do and T. Tran, “Theory and evaluation metrics for learning disentangled representations,” in ICLR, 2020

---

> ### Author Response · Authors · 2024-11-30
>
> * Regarding Theorem 1. Yes, you are right. The backward is only to prove $(x_1,\dots, x_I) \perp (y_1,\dots, y_J)$, not to prove they can be factorized. If $f(x_1, \dots, x_l) \perp g(x_1, \dots, y_J)$, certainly $(x_1,\dots, x_I) \perp g(x_1, \dots, x_J)$, and that's all we want. Please kindly notice that we do not intend to prove $x_1\perp \dots \perp x_I$ or $y_1 \perp \dots \perp y_J$ in Theorem 1.
> * Yes, purely using the original MIG does not effectively assess the concept of group-wise independence, but only single dimensional-wise independence. This is also an important claim in our paper, as stated in Sec. 1 paragraph 3, and Fig. 9(b). To explain the latent $R^2$ measurement in detail, think we generate the data $\boldsymbol x$ using $(z_1, z_2) \perp (z_3, z_4)$, and the algorithm estimated $\hat z_1, \hat z_2, \hat z_3, \hat z_4$. Then, a correct algorithm should be able to find
>     * either $(\hat z_1, \hat z_2)$ aligned with $(z_1, z_2)$ and $(\hat z_3, \hat z_4)$ aligned with $(z_3, z_4)$
>     * or $(\hat z_1, \hat z_2)$ aligned with $(z_3, z_4)$ and $(\hat z_3, \hat z_4)$ aligned with $(z_1, z_2)$
>
>     through affine (linear: rotation, scale, and translation) transformations, and we pick the maximum one as the $R^2$. If an algorithm fails to find this group-wise independence, there will be no appropriate alignment and the resulting latent $R^2$ will be lower. For example, a fully disentangled VAE like $\beta$-TCVAE cannot find group-wise independence, and hence the above two choices all result in bad alignment $R^2$, which finally results in a low latent $R^2$ measurement.
> Except for those supervised metrics, we also have the partial correlation (PC) metric, which directly measures whether $(\hat z_1, \hat z_2) \perp (\hat z_3, \hat z_4)$ from the statistical definition of group-wise independence. See, texts below Eq. 1 and Eq. 5 of our paper. Also please kindly notice that total/partial correlation is just the mutual information, see the [Wikipedia of mutual information](https://en.wikipedia.org/wiki/Mutual_information).
> * The derivation in Appendix 1.3 is very similar to the derivation of the ELBO decomposition in [$\beta$-TCVAE](https://arxiv.org/abs/1802.04942), i.e., Eq. 2 and Appendix C in  [$\beta$-TCVAE](https://arxiv.org/abs/1802.04942). This derivation is for building relationships between $\beta$-TCVAE and our PDisVAE, showing that our PDisVAE is a generalization of their $\beta$-TCVAE, in terms of the ELBO decomposition.
>     * To answer your question, "math" means that line of equation is in the form of KL divergence to explicitly express independence/disentanglement. "code" means how people implement those terms in the code.

---

> > ### Author Response · Authors · 2024-12-03
> > **Any question before the deadline today?**
> >
> > Dear Reviewer 1ysP,
> >
> > If you still have further concerns or want to discuss other questions with us, please let us know before the deadline today (AOE). We will try our best to provide feedback before the author response deadline tomorrow (AOE).
> >
> > Best,
> > Authors

---

### Official Review · Reviewer_xQT7 · 2024-11-07

**Soundness:** 4
**Presentation:** 3
**Contribution:** 3
**Rating:** 5
**Confidence:** 4

**Summary:**

The paper proposes a new type of VAE (Partially Disentangled VAE), which adapts the traditional total correlation (TC) term used in fully disentangled VAEs to a new partial correlation (PC) term. This modification allows the model to capture group-wise independence, offering more flexibility when full independence among latent components is impractical. The authors validate PDisVAE through experiments on synthetic and real-world datasets, demonstrating its effectiveness in achieving interpretable and flexible disentangled representations compared to fully disentangled VAEs. The paper highlights the practical advantages of partial disentanglement, presenting PDisVAE as a robust alternative in complex applications.

**Strengths:**

- 1) The introduction of the partial correlation (PC) term in VAEs is an innovative approach that extends the traditional total correlation (TC), enabling group-wise independence and addressing limitations in full disentanglement.

- 2) The paper is backed by rigorous theoretical derivations and thorough empirical validation on both synthetic and real-world datasets, proving the effectiveness and flexibility of the proposed PDisVAE model.

- 3) By allowing partial disentanglement, PDisVAE broadens the applicability of VAEs in practical scenarios, making it a meaningful advancement for generative modeling and representation learning.

**Weaknesses:**

-1) The paper does not include a dedicated Related Works section, which would help contextualize its contributions and distinguish it from existing literature on disentangled representation learning.  This can greatly improve the presentation as well as make it easier to understand the work in the context of the works before it. Moreover, the citations are very old and the recent works have not been cited appropriately.

-2) The paper does not provide a numerical comparison table for evaluating the performance of PDisVAE against other models on synthetic as well as real world datasets, limiting the clarity and impact of its empirical results. Could the authors provide a table with quantitative comparisons on different datasets to reinforce the empirical validation of PDisVAE? Reporting MIG-SUP scores would also be a plus. Note: The reviewer is willing to improve the scores if the work can be substatiated with quantitative results.

**Questions:**

See Weaknesses

---

> ### Author Response · Authors · 2024-11-21
>
> Dear Reviewer xQT7,
>
> Thank you very much for your time and valuable comments on our paper. We highly appreciate your recognition of the strengths of our paper. Hopefully, the following responses could resolve most of your concerns and answer your questions.
>
> > Related works
>
> Thanks for reminding us. We apologize that there should have been a section of related works. To answer similar questions asked by other reviewers, we have added a section of related works in Appendix A.5 as an early response. We are working on running necessary comparison experiments. We will reorganize our draft once we finish it and send you a notification.
>
> > Numerical comparison
>
> Please kindly notice that rigorous statistical comparisons were already in Fig. 2(b), Fig. 5(b), and Fig. 8 of our initial submission.
>
> Specifically, when we know the true latents (labels), so we compute the $R^2$ metric between the estimated latents and the true as a supervised measurement. The pairwise $t$-test of not only the supervised latent $R^2$ but also the unsupervised partial correlation (PC) in Fig. 2(b), and Fig. 5(b) rigorously show significantly better results from our PDisVAE.
>
> Since MIG is a commonly used metric for full disentanglement, we have tried to adapt it to evaluate partial disentanglement (group-wise independence) and presented a table including both $R^2$ and MIG in Appendix A.6. Specifically, to answer similar comparison questions asked by other reviewers, we have added a comprehensive table comparison in Appendix A.6 to show the superiority of our PDisVAE. Some metrics and interpretations were already presented in Fig. 7 and Fig. 8 of our initial submission.

---

> > ### Author Response · Authors · 2024-11-27
> > **Sincerely looking forward to your feedback before deadline**
> >
> > Dear Reviewer xQT7,
> >
> > Thank you very much again for your initial review of our paper. Given it is close to the discussion deadline, and we are not sure whether you are satisfied with our answers and revised paper or whether there are still some concerns that have not been fully resolved, we would like to kindly remind you that we have answered all your questions and concerns mentioned in the weaknesses. Our latest revision and new materials in the appendix provide a clearer picture of our work. We appreciate your time and invaluable feedback.
> >
> > Best,

---

> > > ### Comment · Reviewer_xQT7 · 2024-12-01
> > > **Thank you for your response**
> > >
> > > Thank you for adding the comparisons with other prior works. I will retain my original scores at this time.
> > >
> > > I have a question regarding figure 9: I observe several unwanted changes in the images generated. For example, for 4 groups, when the authors try to vary background color space, I observe changes in the facial features. This is just one of the examples but I observe unwanted changes in several places. How can one explain this?

---

> ### Author Response · Authors · 2024-12-01
>
> Dear Reviewer xQT7,
>
> This totally makes sense. That means this rank-3 group mainly contains background color but also some other information (like some tiny facial features you observed). Please kindly notice that semantic meaning does not always correspond to an independent component or an independent group. Not only our PDisVAE but also fully disentangled VAEs like $\beta$-TCVAE and FactorVAE are essentially trying to find "statistically" independent groups/components, but an independent component/group might not have one semantic meaning and that depends on reasonable interpretation.
>
> It is possible that a group contains more than one semantic meaning. For example, it is likely that females have more warm backgrounds and males have more cold backgrounds. In this case, the background warm/cold is entangled with gender. In this case, we cannot separate these two semantic meanings since they are statistically dependent/entangled.
>
> In our example of background color, especially Fig. 9(b), we interpret that group as background color based on our human understanding. However, we cannot rigorously prove that the background color is totally independent of the tiny facial feature changes you observed. This is actually an important point we want to stress in this paper, like in Sec. 1 paragraph 3, Fig. 7(a), and Fig. 9(b). We can summarize the following four possibilities:
> * one semantic meaning corresponds to one latent component (fully independent)
> * one semantic meaning corresponds to several entangled latent components (a latent group)
> * several semantic meanings correspond to one latent component (semantic meanings are entangled and encoded by one latent component)
> * several semantic meanings correspond to several latent components (semantic meanings are entangled and encoded by several latent components)
>
> This is the key reason we generalize fully disentangled VAE to partially disentangled VAE (PDisVAE) since PDisVAE considers all these possibilities that exist in nearly all real-world datasets (maybe with the probability of 1). We view this as our paper's key take-home message that we really need to jump out of the stereotype that one latent component should correspond to one semantic meaning.
>
> For example, in the partial dsprites (pdsprites) dataset shown in Fig. 7(a), although we humans think $x$ location and $y$ location are two separable semantic meanings, they are statistically dependent/entangled with each other, so we cannot separate them but put them in one group, and that is why fully disentangled VAEs (e.g., $\beta$-TCVAE) fails with this dataset (Fig. 7(b)). You can think $x$ and $y$ as two semantic meanings or say $(x, y)$ "location" is one semantic meaning, but the ground truth is that $x$ location and $y$ location are entangled, not statistically separable, and hence should be encoded by a latent group of at least rank-2.
>
> A similar reason also holds for the color distribution we plot in Fig. 9(b). If you use a fully disentangled VAE, you can only interpret that the background color (from red to blue, a curve in HSV space) is encoded by one latent component, but that might not be the fact. We do show in Fig. 9(b) that with more latent components entangled with each other as a group, the background color semantic meaning can be expressed more fully (a 2D manifold or a restricted 3D region that is not evenly distributed).
>
> To answer your question again, let us give another example. In Fig. 1(c) of the [$\beta$-TCVAE paper](https://arxiv.org/pdf/1802.04942), you will find that they claimed those latent variations as gender, but background color changes. That means gender and background color (two semantic meanings) are very likely to be entangled with each other, and this is actually reasonable. No one can promise an absolutely perfect correspondence between semantic meaning(s) and a latent component/group. All researchers can do is validate the correctness of their method on synthetic dataset, like we do in Sec. 4.1 and Sec. 4.2, and to get more interpretable (but cannot promise perfect correspondence) disentanglement results on real-world datasets. Generally speaking, it is nearly impossible for all kinds of disentangling methods to find pure correspondence between a latent component/group and one semantic meaning on real-world datasets. At least there are some noises including other semantic meanings of tiny magnitude. This kind of result should be acceptable in the field of representational learning (disentanglement), especially on real-world datasets where there is no true latent. Otherwise, any interpretation from any method could have small flaws (that can even come from random seeds or the floating point precision of the training device).
>
> If Reviewer xQT7 still has more questions on this point, we'd love to provide more detailed discussions on it.
>
> Best,
> Authors

---

> > ### Author Response · Authors · 2024-12-03
> > **Any question before the deadline today?**
> >
> > Dear Reviewer xQT7,
> >
> > If you still have further concerns or want to discuss other questions with us, please let us know before the deadline today (AOE). We will try our best to provide feedback before the author response deadline tomorrow (AOE).
> >
> > Best,
> > Authors

---

### Official Review · Reviewer_AfSM · 2024-11-08

**Soundness:** 3
**Presentation:** 3
**Contribution:** 2
**Rating:** 6
**Confidence:** 5

**Summary:**

This paper describes a modification of the loss function of a variational autoencoder which results in a group-wise independent latent distribution. Similar to the existing TC-beta-VAE (Chen et al 2018), where the evidence lower bound is modified to contain a total correlation term that results in independence of the latent components, in this work the total correlation is computed group-wise, resulting in latent representations in which several groups of latent components become independent.

**Strengths:**

The paper is well written and with compelling experiments that demonstrate the effectiveness of the proposed modification of the loss function for training a VAE with group-wise independent latent distribution. An importance sampling method for estimating the group-wise posterior lower variance than the one in Chen et al 2018 is proposed.

Especially the three cases, non-separable dependent, rank-deficient, and independent presented in section 3.3 are tested with a dedicated experiment on synthetic data that demonstrate the effectiveness of the proposed method.

**Weaknesses:**

Novelty seems to be limited because the idea of group-wise independence is not novel. A prior that results in a group-wise independent latent distribution was already proposed in [1]. In contrast that method does not need to compute a group-wise posterior which simplifies the overall training process and removes the necessity of importance sampling.

Technical remarks:
The Pdf-file takes very long to render in Adobe - perhaps this can be solved by sparsifying the scatter plots.


[1] Stühmer, J., Turner, R., Nowozin, S. (2020). Independent subspace analysis for unsupervised learning of disentangled representations. In International Conference on Artificial Intelligence and Statistics (pp. 1200-1210). PMLR.

**Questions:**

Please discuss the work [1] in section 2.2 as additional VAE with a non-gaussian prior and its relation to ICA, and additionally compare to [1] as related baseline with group-wise independent latent prior in your experiments.

So far only dsprites, celebA and a dataset from neuroscience seem to have been used for evaluation. Another good test case would be to integrate the proposed method into disentanglement lib [2] which would enable the direct comparison to the baselines and datasets evaluated in [3], e.g. shapes3d and cars3d.

[2] https://github.com/google-research/disentanglement_lib
[3] Locatello, F., Bauer, S., Lucic, M., Raetsch, G., Gelly, S., Schölkopf, B. and Bachem, O., 2019, May. Challenging common assumptions in the unsupervised learning of disentangled representations. In international conference on machine learning (pp. 4114-4124). PMLR.

---

> ### Author Response · Authors · 2024-11-22
>
> Dear Reviewer AfSM,
>
> Thank you very much for your time and valuable comments on our paper. We highly appreciate your recognition of the strengths of our paper. Hopefully, the following responses could resolve most of your concerns and answer your questions.
>
> > Comparison with Stühmer, J., Turner, R., Nowozin, S. (2020). Independent subspace analysis for unsupervised learning of disentangled representations. In International Conference on Artificial Intelligence and Statistics (pp. 1200-1210). PMLR.
>
> Thanks for reminding us of this paper. We appreciate this nice suggestion. We have added a section of related works to Appendix A.5. The comparison with the aforementioned paper and other methods has been added to Tab. 4 in Appendix A.6. We will reorganize our draft once we finish them and send you a notification. For your convenience, we show the comparison results here:
>
> The table below presents the partial correlation (PC), latent $R^2$, MSE, and mutual information gap (MIG) evaluated for the partial disentanglement (group-wise independence) in the estimated latent.
>
> |                | PC $\downarrow$      | $R^2$ $\uparrow$     | MSE $\downarrow$     | MIG $\uparrow$       |
> |----------------|----------------------|----------------------|----------------------|----------------------|
> | VAE            | 1.01 (0.02)          | 0.22 (0.04)          | 0.29 (0.02)          | 0.15 (0.10)          |
> | ICA            | 1.76 (0.07)          | 0.22 (0.06)          | 0.28 (0.03)          | 0.14 (0.09)          |
> | ISA            | **0.70 (0.01)** | 0.23 (0.02)          | 0.33 (0.01)          | 0.24 (0.08)          |
> | $\beta$-TCVAE  | 0.91 (0.10)          | 0.33 (0.06)          | **0.24 (0.04)** | 0.36 (0.13)          |
> | $\alpha$-TCVAE | 1.84 (0.03)          | 0.31 (0.02)          | 0.27 (0.01)          | 0.29 (0.09)          |
> | PDisVAE        | **0.68 (0.04)** | **0.54 (0.08)** | **0.23 (0.04)** | **0.49 (0.07)** |
>
> Since only ISA and our PDisVAE explicitly require group-wise independence rather than strong full independence, only ISA and our PDisVAE obtain the lowest PC. Compared with ISA, PDisVAE is more flexible since we achieve partial disentanglement by adding a PC penalty term to the loss function. Via the PC penalty, the latent distribution is estimated through the aggregated posterior $q(\boldsymbol z)$ from the learned decoder, rather than a fixed $L^p$-nested prior in ISA. Therefore, PDisVAE obtains more accurate and partially disentangled latent when evaluated with the true latent (labels).
>
> Although the ISA paper proposed the group-wise independence method through the $L^p$-nested distribution, they did not conduct experiments on synthetic datasets with truly partially disentangled latent. Their application on the dsprites dataset is also a fully disentangled dataset. Compared to their ISA method, therefore, we have comprehensive experiments on both synthetic and real-world datasets to validate and show the existence of partially disentangled latent in many disentangling problems and show the effectiveness of our method in finding those group-wise independent latents.
>
> > PDF takes long to render.
>
> We have changed those large PDF figures to PNG figures and the issue was resolved.
>
> > Integrate into disentanglement library.
>
> Due to the current time limit and unpublished situation of this draft, we plan and would like to incorporate this method into the disentanglement library in the future once it is accepted.

---

> > ### Author Response · Authors · 2024-11-27
> > **Sincerely looking forward to your feedback before deadline**
> >
> > Dear Reviewer AfSM,
> >
> > Thank you very much again for your initial review of our paper. Given it is close to the discussion deadline, and we are not sure whether you are satisfied with our answers and revised paper or whether there are still some concerns that have not been fully resolved, we would like to kindly remind you that we have answered all your questions and concerns mentioned in the weaknesses. Our latest revision and new materials in the appendix provide a clearer picture of our work. We appreciate your time and invaluable feedback.
> >
> > Best,

---

> > > ### Comment · Reviewer_AfSM · 2024-11-28
> > >
> > > I thank the authors for their reply and for performing additional experiments. This definitely has strengthened the paper.
> > > There seems to be some further points with other reviewers that need clarification, so I will keep my score for now.
> > > At this time, I do no have any further questions to the authors and look forward to the reviewer discussion.

---

### Author Response · Authors · 2024-11-23
**Modified version**

Dear all reviewers,

According to all your invaluable suggestions, we have modified our paper to include more related work discussions, comparison results, and so on. The modified main content is highlighted. New (sub)sections A.3.3, A.5, and A.6 were also added to the Appendix.

---

### Meta-Review · Area_Chair_t9ei · 2024-12-21

**Metareview:**

The paper proposes the Partially Disentangled VAE (PDisVAE), a novel extension of VAEs that replaces the traditional total correlation (TC) term used in fully disentangled VAEs with a partial correlation (PC) term. This adjustment enables the model to achieve group-wise independence among latent variables, offering greater flexibility in scenarios where full independence is impractical.

While the introduction of the PC term provides a new perspective on group-wise independence, the reviewers identified several shortcomings. The paper lacks a dedicated Related Works section to position its contributions within the broader literature, and the references include mostly older studies, omitting relevant recent works. Furthermore, the idea of group-wise independence is not entirely novel, as similar approaches, such as those proposed by Stühmer et al. (2020), have been shown to achieve group-wise independent latent distributions without the need for complex group-wise posterior computations or importance sampling. Concerns were also raised about the assumptions and proof of Theorem 1, which require further substantiation.

**Additional Comments On Reviewer Discussion:**

The novelty of the proposed approach appears limited, as the concept of group-wise independence is not entirely new. A prior achieving group-wise independent latent distributions was previously introduced in Stühmer et al. (2020), which notably does not require computing a group-wise posterior. This simplifies the training process and eliminates the need for importance sampling.

During the rebuttal phase, the authors provided a detailed response and conducted additional experiments, which have improved the paper to some extent. However, certain points raised by other reviewers remain unclear and require further clarification. Overall, while the paper introduces incremental improvements, its innovation is constrained and does not provide substantial new insights.

---

### Decision · Program_Chairs · 2025-01-22

Reject